# Dopaminergic and opioidergic regulation during anticipation and consumption of social and nonsocial rewards

**Sebastian Korb[1,2]\*, Sebastian J Götzendorfer[2], Claudia Massaccesi[3], Patrick Sezen[4], Irene Graf[4], Matthäus Willeit[4], Christoph Eisenegger[2], Giorgia Silani[3]\***

[1]Department of Psychology, University of Essex, Colchester, United Kingdom; [2]Department of Cognition, Emotion and Methods in Psychology, University of Vienna, Vienna, Austria; [3]Department of Clinical and Health Psychology, University of Vienna, Vienna, Austria; [4]Department of Psychiatry and Psychotherapy, Medical University of Vienna, Vienna, Austria

**Abstract** The observation of animal orofacial and behavioral reactions has played a fundamental role in research on reward but is seldom assessed in humans. Healthy volunteers (N = 131) received 400 mg of the dopaminergic antagonist amisulpride, 50 mg of the opioidergic antagonist naltrexone, or placebo. Subjective ratings, physical effort, and facial reactions to matched primary social (affective touch) and nonsocial (food) rewards were assessed. Both drugs resulted in lower physical effort and greater negative facial reactions during reward anticipation, especially of food rewards. Only opioidergic manipulation through naltrexone led to a reduction in positive facial reactions to liked rewards during reward consumption. Subjective ratings of wanting and liking were not modulated by either drug. Results suggest that facial reactions during anticipated and experienced pleasure rely on partly different neurochemical systems, and also that the neurochemical bases for food and touch rewards are not identical.

**\*For correspondence:**
sebastian.korb@essex.ac.uk (SK);
giorgia.silani@univie.ac.at (GS)

**Competing interests:** The authors declare that no competing interests exist.

## Introduction

Rewards are salient stimuli, objects, events, and situations that induce approach and consummatory behavior by their intrinsic relevance for survival or because experience has taught us that they are pleasurable (*Schultz, 2015*). Today, our understanding of the neurochemical basis of reward processing rests on 30 years of animal research, and on preliminary confirmatory findings in humans, which led to the identification of two distinct components: *wanting*, that is, the motivation to mobilize effort to obtain a reward, and *liking*, that is, the hedonic experience evoked by its consumption (*Berridge, 1996*; *Berridge, 2018*; *Berridge and Kringelbach, 2015*; *Berridge and Robinson, 1998*). This conceptual division is paralleled in cognitive theories of economic decision making (*Kahneman et al., 1997*; *Berridge and O'Doherty, 2014*) that similarly distinguish between *decision utility* (how much the value attached to an outcome determines its choice or pursuit) and *experienced utility* (referring to the hedonic experience generated by an outcome).

In animal research, the 'taste reactivity test', a method to assess eating-related pleasure by observing facial and bodily reactions of animals and human infants to palatable and aversive tastes, has played a fundamental role in the identification of discrete reward systems in the brain (*Barbano and Cador, 2007*; *Berridge, 2000*; *Dolensek et al., 2020*; *Grill and Norgren, 1978*; *Steiner et al., 2001*). Indeed, it has been shown that neither pharmacological disruption nor extensive lesion of dopaminergic neurons affects facial liking reactions (e.g. relaxed facial muscles and licking of the lips) to the consumption of sweet foods in rats (*Berridge et al., 1989*; *Treit and Berridge,*

**eLife digest** Studies in rats and other species have shown that two chemical messengers in the brain regulate how much an animal desires a reward, and how pleasant receiving the reward is. In this context, chemicals called opioids control both wanting and enjoying a reward, whereas a chemical called dopamine only regulates how much an animal desires it. However, since these results were obtained from research performed on animals, further studies are needed to determine if these chemicals play the same roles in the human brain.

Korb et al. show that the same brain chemicals that control reward anticipation and pleasure in rats are also at work in humans. In the experiment, 131 healthy volunteers received either a drug that blocks opioid signaling in the brain, a drug that blocks dopamine signaling, or a placebo, a pill with no effect. Then, participants were given, on several occasions, either sweet milk with chocolate or a gentle caress on the forearm. Participants rated how much they wanted each of the rewards before receiving it, and how much they liked it after experiencing it. To measure their implicit wanting of the reward, participants also pressed a force-measuring device to increase their chances of receiving the reward. Additionally, small electrodes measured the movement of the volunteer's smiling or frowning muscles to detect changes in facial expressions of pleasure.

Volunteers taking either drug pressed on the device less hard than the participants taking the placebo, suggesting they did not want the rewards as much, and they frowned more as they anticipated the reward, indicating less anticipatory pleasure. However, only the volunteers taking the opioid-blocking drug smiled less when they received a reward, indicating that these participants did not get as much pleasure as others out of receiving it. These differences were most pronounced when volunteers looked at or received the sweet milk with chocolate.

This experiment helps to shed light on the chemicals in the human brain that are involved in reward-seeking behaviors. In the future, the results may be useful for developing better treatments for addictions.

---

*1990*), and that greater mesolimbic dopamine release induced by electric stimulation of the hypothalamus results in greater food intake without modulating hedonic reactions (*Berridge and Valenstein, 1991*). On the other hand, (facial) hedonic reactions to sensory pleasure are amplified by opioid, orexin, and endocannabinoid stimulation of various 'hedonic hotspots' of the brain, including the nucleus accumbens (NAc) shell and limbic areas such as the insula and the orbitofrontal cortex (*Berridge and Kringelbach, 2015*). These stimulations not only increase liking but also result in food approach and feeding behavior (*Peciña and Smith, 2010*; *Taha, 2010*).

Evidence of similar neurochemical parsing of reward processing in humans is mainly derived from research in clinical populations and a handful of recent pharmacological studies in healthy volunteers. For example, stimulation of D2/D3 receptors through dopamine agonists can induce compulsive medicament use, gambling, shopping, hypersexuality, and other addictive activities in some patients with Parkinson's disease, often without corresponding changes in subjective liking (*Callesen et al., 2013*; *Evans et al., 2006*; *Weintraub et al., 2010*; but see *Meyer et al., 2019* for an account of the complexity of compulsive disorders in Parkinson's disease). Evidence for disrupted motivation to gain immediate rewards has been observed after dopamine D2/D3 receptors blockade in healthy volunteers, in both a pavlovian-instrumental-transfer task and a delay discounting task (*Weber et al., 2016*). Administration of μ-opioid receptor agonists in healthy individuals has been associated with changes in subjective feelings and motivational responses to different types of rewards, as indicated, for example, by higher pleasantness of the highest-calorie (but least palatable) food option available (*Eikemo et al., 2016*), greater effort to view, and liking of the most attractive opposite-sex faces (*Chelnokova et al., 2014*), stronger preference for stimuli with high-reward probability (*Eikemo et al., 2017*), and enhanced emotional ratings to positive and negative images (*Atlas et al., 2014*). Furthermore, administration of the non-selective opioid receptor antagonist naloxone to healthy men decreased subjective pleasure associated with viewing erotic pictures and reduced the activation of reward related brain regions such as the ventral striatum (*Buchel et al., 2018*).

Despite the progress made, the animal research is only partly informative to comprehend reward processing in humans. While animal research allows to investigate the activity of neurons and neurotransmitters in a much more targeted way, it is also limited to certain measures of liking (i.e. behavior and facial expressions, while humans can also provide subjective reports), and has mainly focused on food rewards in the past. Moreover, human and animal research about the neurochemical regulation of reward processing remain difficult to compare, as human pharmacological studies have struggled to adopt translational paradigms and an operationalization of reward that resembles the one used in animal research, that is, measuring decision utility and experienced utility in the same task, providing primary rewards on a trial-by-trial basis, and/or using objective hedonic reactions to consumed rewards, in addition to relying on subjective verbal report (*Der-Avakian et al., 2016*; *Pool et al., 2016*).

Including the recording of hedonic facial reactions, the 'gold standard' in animal research on the neurochemical regulation of reward processing in human studies seems to be a promising avenue in this regard. Recently, the use of facial electromyography (fEMG) has gained increased attention in the context of human reward processing. Results suggested that human adults relax the corrugator muscle (involved in frowning), and to a lesser extent activate the zygomaticus muscle (involved in smiling), during both anticipation and consumption of different types of pleasurable stimuli, although differences between types of rewards exist (*Bershad et al., 2019*; *Franzen and Brinkmann, 2016*; *Korb et al., 2020*; *Mayo et al., 2018*; *Pawling et al., 2017*; *Rasch et al., 2015*; *Ree et al., 2019*; *Sato et al., 2020*; *Wu et al., 2015*). Notably, to the best of our knowledge, no study has yet investigated implicit hedonic facial reactions to different types of rewards after pharmacological drug challenge in humans.

To fill this knowledge gap, we pharmacologically manipulated the dopaminergic and opioidergic systems in humans via oral administration of the highly selective D2/D3 dopamine receptor antagonist amisulpride (400 mg), the non-selective opioid receptor antagonist naltrexone (50 mg), or placebo, in a randomized, double-blind, between-subject design in 131 healthy volunteers (group sizes were 42, 44, and 45, respectively, for amisulpride, naltrexone, and placebo), and investigated the effects with a recently developed experimental paradigm (*Korb et al., 2020*), in which reward processing is operationalized similarly to animal research. Explicit subjective ratings of wanting and liking, physical effort (squeezing of an individually thresholded hand-dynamometer to obtain rewards) and implicit hedonic reactions (fEMG) during *anticipation* and *consumption* of primary social and nonsocial rewards of similar magnitude were obtained on a trial-by-trial basis (*Figure 1*). Sweet milk with different concentrations of chocolate flavour served as nonsocial food rewards. Gentle caresses to the forearm, delivered by a same-sex experimenter at different speeds, resulting in different levels of pleasantness (*Ackerley et al., 2014*; *Löken et al., 2009*; *McGlone et al., 2014*), served as nonsexual social rewards.

By adopting a translational approach, which makes human research comparable to animal research (e.g. measuring both real effort and hedonic facial reactions to primary rewards), we investigated two fundamental yet unresolved research questions: (1) to what extent do motivational and hedonic implicit and explicit responses during the *anticipation* and *consumption* of rewards rely on the dopaminergic and opioidergic systems in humans, and (2) do food and touch rewards share the same neurochemical basis in humans?

We made the following hypotheses based on the literature. First, because liking relies heavily on the opioidergic but not the dopaminergic system (*Berridge and Kringelbach, 2015*), subjective ratings of liking, and hedonic facial reactions during reward *consumption*, were expected to be lower after administration of the opioid antagonist naltrexone, compared to placebo, but not after administration of the dopamine antagonist amisulpride, particularly for the most preferred rewards (*Eikemo et al., 2016*; *Smith and Berridge, 2007*). Second, because wanting is believed to be regulated by the dopaminergic and opioidergic systems (*Peciña and Berridge, 2013*), we expected subjective ratings of wanting, and physical effort applied to obtain the preferred announced reward, to be lower after administration of both naltrexone and the D2/D3 receptor antagonist amisulpride. Third, because facial responses during reward *anticipation* – previously shown to occur to learned cues for rewards in rats (*Delamater et al., 1986*), and humans (*Korb et al., 2020*) – may reflect anticipated pleasure during a period commonly associated with wanting, they were expected to be affected by naltrexone, as well as by amisulpride, compared to placebo. Finally, based on fEMG results showing similar hedonic facial reactions to food and touch rewards, such as relaxation of the

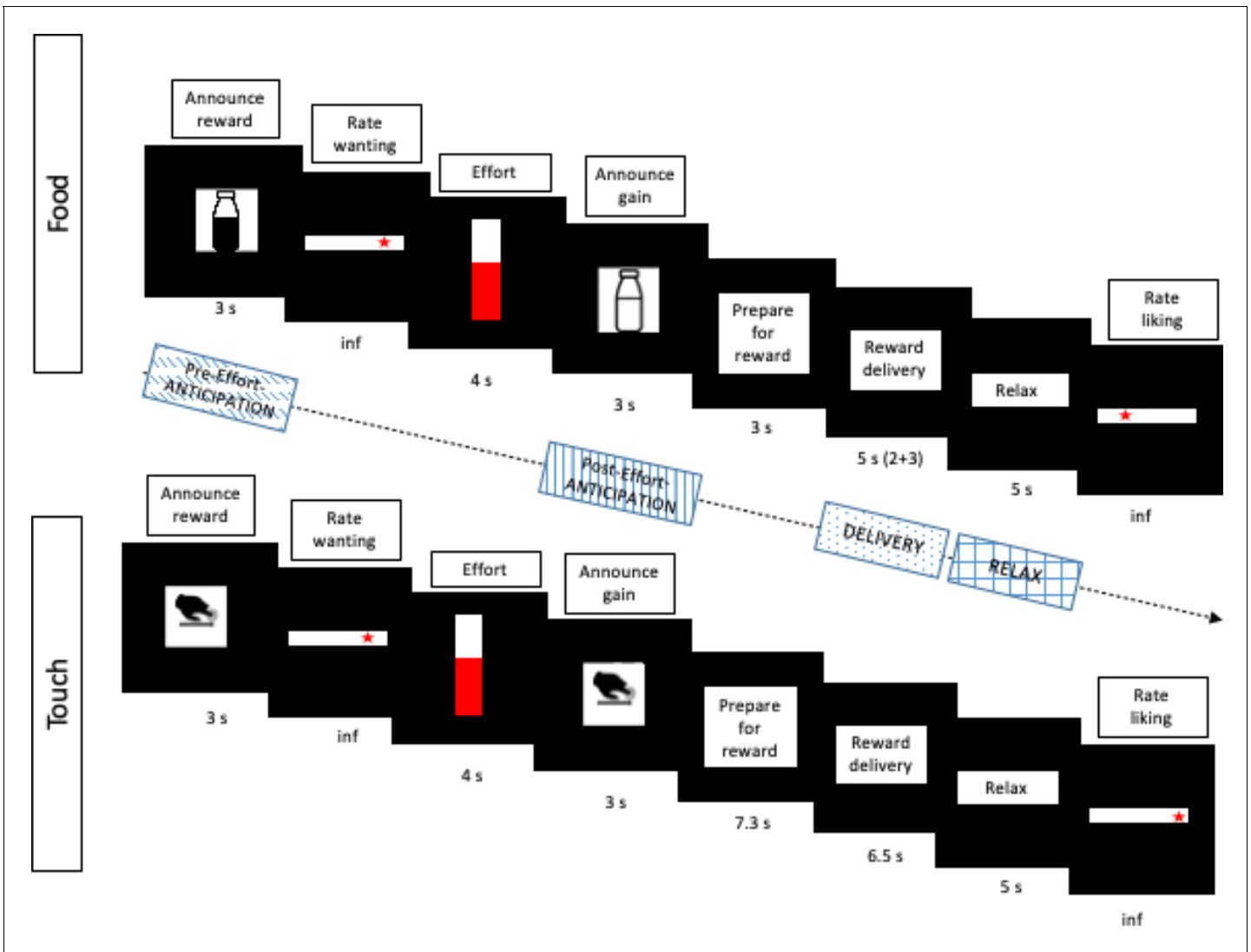

**Figure 1.** Main elements in each trial for the food (top) and touch (bottom) reward types. Before the main task, participants experienced and ranked three food stimuli and three touch stimuli, based on liking (*Figure 2*). In the main task (here depicted), the highest-ranked ('high') reward was announced in half of the trials and the second-highest ranked ('low') reward was announced in the other half of trials. The probability of obtaining the announced reward was determined linearly by participants' hand-squeezing effort, which was indicated in real-time. Participants knew that they would obtain the announced reward if they reached the top of the displayed vertical bar, which corresponded to their previously measured maximum voluntary contraction (MVC). The gained reward (which was either the one announced at the beginning of the trial, or – in the case of lower probability due to less squeezing – the least-liked 'verylow' reward) was then announced and delivered. To assess reward anticipation, EMG data was analyzed during the Pre-Effort anticipation period (3 s) at the beginning of the trial, when a possible reward was announced, as well as during the Post-Effort anticipation period (3 s announcement) preceding reward delivery. To investigate reward consumption, EMG data was analyzed during reward Delivery (5 s for food and 6.5 s for touch), and in the immediately following Relax phase (5 s). Rating slides stayed on screen indefinitely, or until participants' button press. For a representation of all trial elements see *Figure 1—figure supplement 1*.

The online version of this article includes the following figure supplement(s) for figure 1:

**Figure supplement 1.** All trial elements.

corrugator supercilii muscle and in some cases activation of the zygomaticus major muscle (*Bershad et al., 2019*; *Korb et al., 2020*; *Mayo et al., 2018*; *Pawling et al., 2017*; *Ree et al., 2019*; *Sato et al., 2020*), and on evidence from neuroimaging studies that supports the 'common currency hypothesis' of reward processing (*Berridge and Kringelbach, 2015*; *Ruff and Fehr, 2014*), we expected the same pattern of results for both types of rewards.

## Results

### Matching of drug groups

In order to rule out eventual group differences that were not of interest, we conducted a series of statistical tests to verify the matching of the three groups.

The three drug groups did not differ significantly in rankings of rewards *before* the main task, as shown by the absence of a significant Drug X Reward Level interaction for both food and touch rewards (*Figure 2*). Only a significant main effect of Reward Level was found for food ($X^2$ (2)=78.1, p<0.001) and for touch rewards ($X^2$ (2)=115.71, p<0.001), confirming the expected pattern of preferred food rewards (milk with greater chocolate content being preferred to milk with lower chocolate content), and of touch rewards (slower caresses being preferred to faster caresses).

In the main task, the level of reward (high, low, verylow) received in each trial depended on both the announcement cue at the beginning (high or low) and the force exerted to obtain it (verylow rewards were only obtained when participants exerted low effort, which linearly converted into low

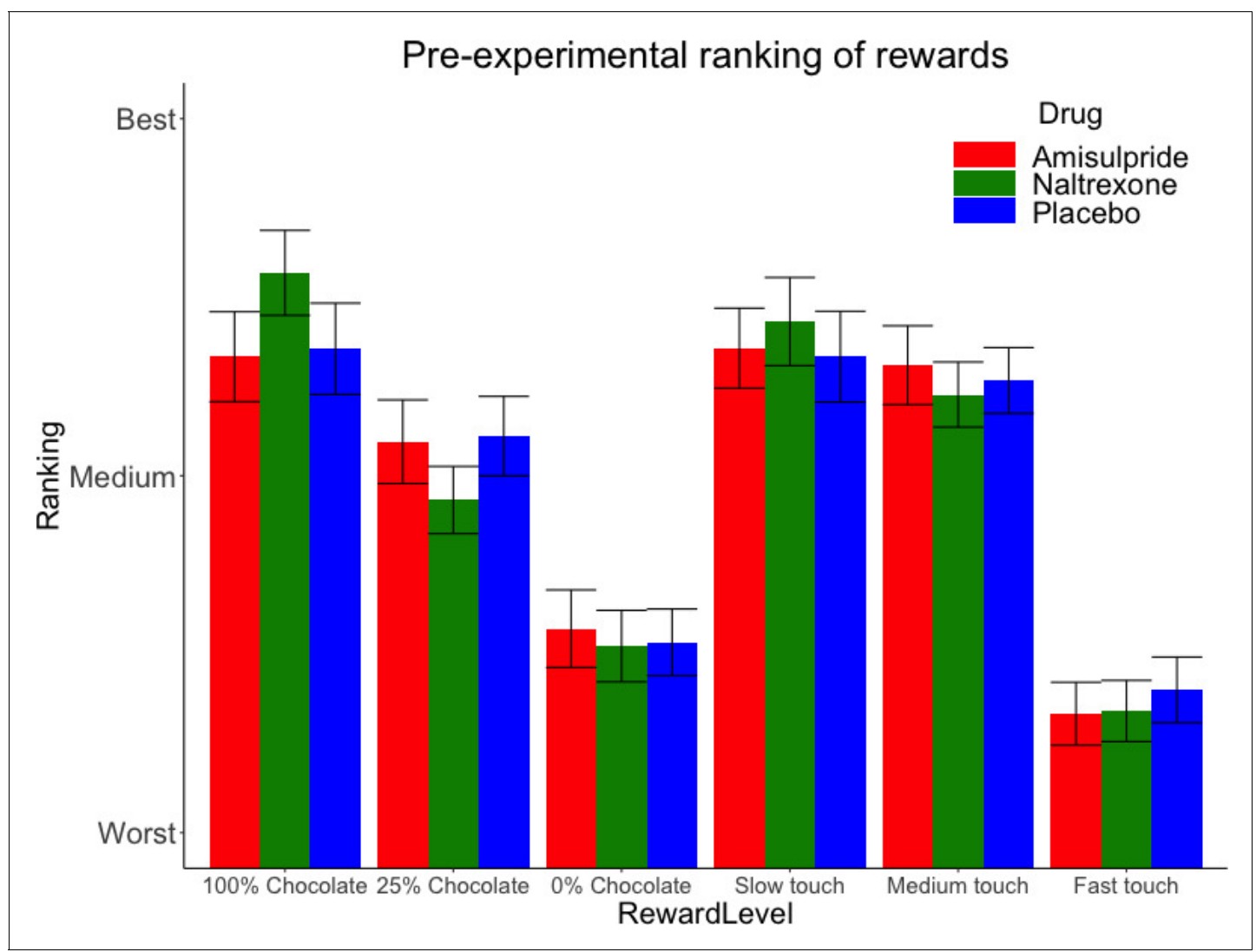

**Figure 2.** Pre-experimental mean (SE) rankings of rewards by preference. This ranking occurred just before the main task, which adapted to these preferences by using for each Reward Type (food, touch) the highest ranked stimulus as 'high' reward, the second-highest ranked stimulus as 'low' reward, and the lowest-ranked stimulus as 'verylow' reward.

The online version of this article includes the following figure supplement(s) for figure 2:

**Figure supplement 1.** Effect of time (trial) on ratings and effort.

probability to obtain the announced reward). The number of trials in which high, low, and verylow rewards were obtained did not differ significantly across groups. Only a significant main effect of Reward Level was found ($F(2, 763)=27.84$, p<0.001), due to a greater number of high ($M = 33.07$, $SD = 4.97$) than low ($M = 29.85$, $SD = 6.03$) and verylow ($M = 17.14$, $SD = 8.99$) trials received, across all three drug groups and both reward types.

As expected, ratings of wanting and liking, as well as effort exerted to obtained the cued rewards, decreased over the course of the experiment due to habituation and fatigue (*Figure 2— figure supplement 1*). This decrease was similar across drug groups. Nevertheless, the covariate Block (recoded into first and second block separately by Reward Type) was included in the analyses of behavioral and fEMG measures (see below), to control for habituation and fatigue.

The three groups of participants did not differ significantly in their maximum voluntary contraction (MVC) of the hand dynamometer, which was measured right before the main task and at the end of the main task, nor in their positive and negative mood measured with the PANAS at time of pill intake and 3 hr later (all $b < 0.6$, all $t < 0.8$, all p>0.4; see *Table 1*). Finally, the three groups of participants did not differ significantly in terms of possible side effects, which were self-reported at time of pill intake and 3 hr later (see *Table 1* for nausea scores).

## Explicit measures: ratings of wanting, ratings of liking, and physical effort

Drug effects were investigated on ratings of wanting provided at the beginning of each trial, on physical effort to obtain an announced reward, and on ratings of liking provided after having obtained a reward. Interactions with the factor Drug were only found for physical effort.

Behavioral analyses on ratings of wanting (*Figure 3—figure supplement 1, A–B*) resulted in an expected significant main effect of Reward Level ($F(1, 128.0)=119.28$, p<0.001), due to higher ratings of wanting for high reward ($M = 4.83$, $SD = 4.31$) compared to low reward ($M = 1.14$, $SD = 4.46$), and a significant main effect of Block ($F(1, 9653.7)=54.30$, p<0.001) due to decreasing wanting from the first ($M = 3.19$, $SD = 4.70$) to the second block ($M = 2.85$, $SD = 4.83$). All other effects were not significant (all $F < 2.3$, all p>0.11).

To verify the lack of drug effects on ratings of wanting, we ran the same LMM using the full Bayesian method with the brms package (*Bürkner, 2017*). A normal prior with M = 0 and SD = 5 was defined for population-level (fixed) effects, and a half student-t prior with 3 degrees of freedom, M = 0, and scaling parameter = 4.7, was set for the standard deviation of subject-specific (random) effects. Results showed that neither amisulpride ($\beta_{mean} = 1$, 95% Bayesian credible interval [−0.03, 2.07]), nor naltrexone ($\beta_{mean} = 0.07$, 95% Bayesian credible interval [−0.94, 1.09]) had credible main effects on ratings of wanting (based on their respective 95% Bayesian credible interval crossing zero), nor did they interact with Reward Level or Reward Type (all $\beta_{mean} < 0.3$, all 95% Bayesian

**Table 1.** Participants' characteristics across groups, as tested with linear regression (the t and p value refer to the main effect of Group).
BMI = Body Mass Index; MVC = Maximum Voluntary Contraction; PANAS = Positive and Negative Affective Schedule; M = Mean; SD = Standard deviation.

|  | Amisulpride | Naltrexone | Placebo | Group differences |
|---|---|---|---|---|
| N (male, female) | 42 (14, 28) | 44 (14, 30) | 45 (15, 30) |  |
| Age M (SD) | 23.7 (4.1) | 22.9 (2.8) | 23.1 (3.7) | $t = -0.73$, p=0.46 |
| BMI M (SD) | 22.7 (2.5) | 23.0 (2.3) | 22.2 (2.5) | $t = -0.99$, p=0.32 |
| MVC M (SD) | 211.9 (86.3) | 208.7 (81.8) | 215.3 (73.1) | $t = 0.19$, p=0.85 |
| PANAS pos T1 M (SD) | 30.5 (5.4) | 29.7 (7.3) | 29.4 (6.7) | $t = -0.8$, p=0.42 |
| PANAS neg T1 M (SD) | 12.1 (3.2) | 14.3 (7.5) | 11.5 (2.1) | $t = -0.7$, p=0.52 |
| PANAS pos T2 M (SD) | 27.1 (6.3) | 24.7 (8.0) | 26.7 (7.4) | $t = -0.3$, p=0.80 |
| PANAS neg T2 M (SD) | 10.1 (2.8) | 12.1 (5.5) | 10.5 (0.9) | $t = -0.5$, p=0.58 |
| Nausea T1 M (SD) | 1.05 (0.2) | 1.02 (0.1) | 1.00 (0.0) | $t = -1.5$, p=0.14 |
| Nausea T2 M (SD) | 1.00 (0.0) | 1.20 (0.6) | 1.00 (0.0) | $t = -0.1$, p=0.93 |

credible interval crossing zero). In addition, a second Bayesian LMM was fitted without the main and interaction effects for the predictor Drug. The two models were compared using a Bayesian leave-one-out cross-validation (LOO-CV; *Vehtari et al., 2017*). This revealed a greater predictive ability for the null model (weight = 0.67, averaging via stacking of predictive distributions) than the full model (weight = 0.33), meaning that the null model is expected to be two times more accurate in predicting new data. One can thus conclude, that taking into account the drug administered does not improve the ability to predict participants' ratings of wanting.

The LMM on effort (*Figure 3—figure supplement 1, C–D*) resulted in the expected significant main effect of Reward Level ($F(1, 128.5)=54.41$, p<0.001), due to stronger force applied for high ($M = 80.49$, $SD = 22.35$) than low rewards ($M = 71.74$, $SD = 25.42$); a significant main effect of Block ($F(1, 7527.4)=175.49$, p<0.001) due to decreasing effort from the first ($M = 78.27$, $SD = 23.79$) to the second block ($M = 74.02$, $SD = 24.65$); and a significant Reward Type X Drug interaction ($F(2, 128.4)=4.71$, p=0.01; *Figure 3A*) reflecting lower effort for food in the amisulpride ($M = 74.98$, $SD = 26.57$) and naltrexone ($M = 73.51$, $SD = 24.43$) groups compared to the placebo ($M = 80.20$,

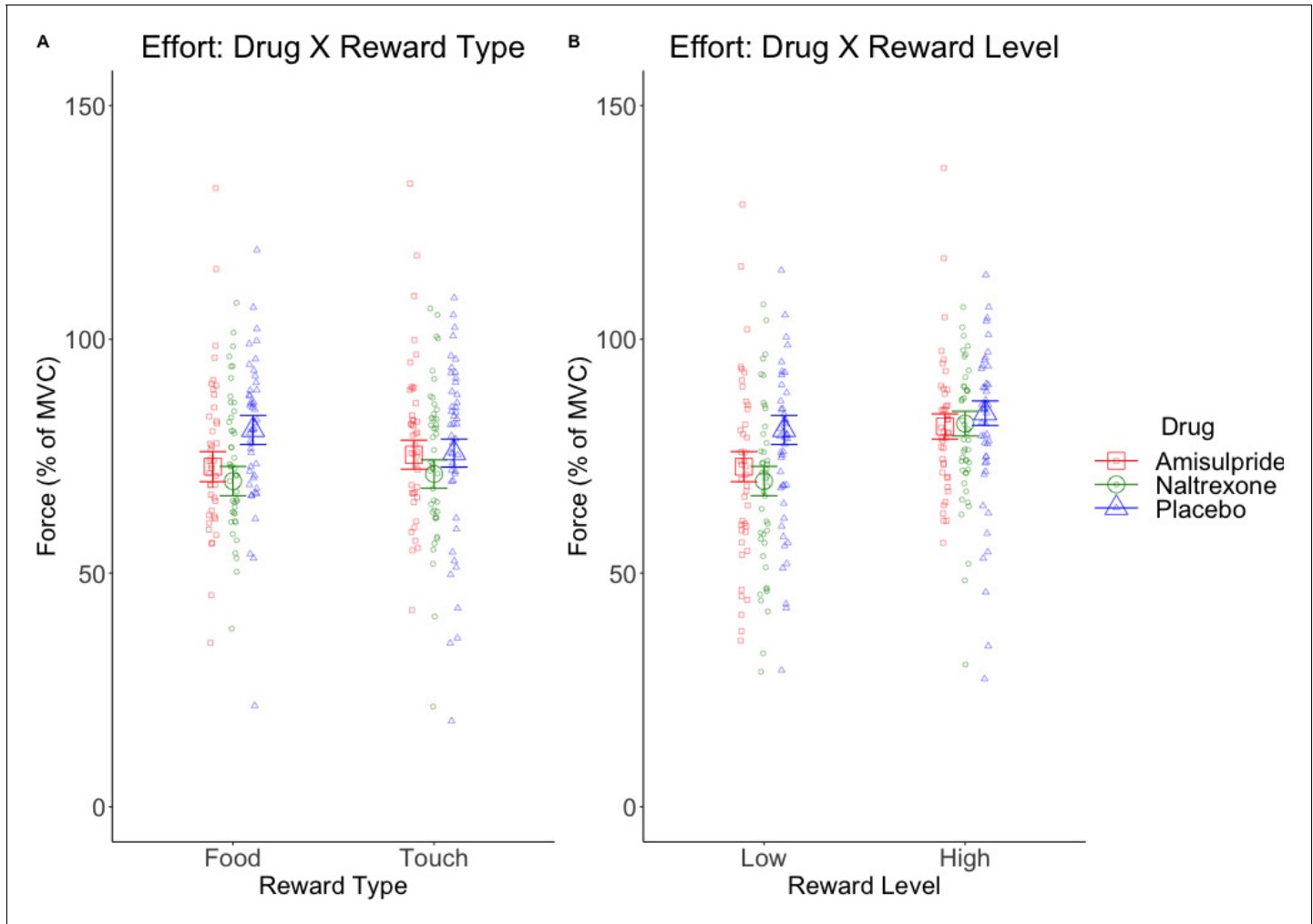

**Figure 3.** Marginal means (and 95% CIs) for interactions with the predictor Drug in behavioral analyses. Physical effort was lower in the amisulpride and naltrexone groups compared to placebo (A) for food but not touch rewards, and (B) non-significantly (p=0.056) for low but not high rewards. This suggests lower wanting after inhibition of both the dopaminergic and the opioidergic systems, specifically for high and low food rewards and for low-level rewards of both reward types. These null effects were confirmed with Bayesian analyses. See *Figure 3—figure supplement 1* for all behavioral results.

The online version of this article includes the following figure supplement(s) for figure 3:

**Figure supplement 1.** Ratings and Effort by Reward Type, Reward Level, and Drug.

SD = 22.41) group, but similar force across drug groups for touch (amisulpride: M = 78.34, SD = 25.14; naltrexone: M = 73.78, SD = 23.15; placebo: M = 76.11, SD = 23.51). The Reward Level X Drug interaction (*Figure 3B*) reflected lower effort for low rewards in the amisulpride (M = 71.67, SD = 27.60) and naltrexone (M = 67.90, SD = 24.45) groups compared to Placebo (M = 75.65, SD = 23.60), but failed to reach significance ($F(2, 128.5)=2.95$, p=0.056). All other effects were not significant (all $F < 0.9$, all p>0.4).

The same LMM on ratings of liking (*Figure 3—figure supplement 1, E–F*) resulted in the main effect of Reward Level ($F(1, 126.4)=150.55$, p<0.001), with greatest liking of high rewards (M = 5.02, SD = 4.10), followed by low rewards (M = 1.79, SD = 4.15), and verylow rewards at the bottom (M = −1.19, SD = 3.89). Decrease of liking over time was shown by a significant main effect of Block ($F(1, 9400.7)=129.40$, p<0.001), due to a decrease in liking from the first (M = 2.90, SD = 4.61) to the second block (M = 2.25, SD = 4.79). All other effects were not significant (all $F < 1.1$, all p>0.34).

The lack of an effect of drug on liking was confirmed by fitting a Bayesian LMM (the same priors were set as for the models on wanting). Neither amisulpride ($\beta_{mean} = 0.78$, 95% Bayesian credible interval [−0.20, 1.77]), nor naltrexone ($\beta_{mean} = 0.25$, 95% Bayesian credible interval [−0.71, 1.18]) had main effects on ratings of liking, nor did they interact with Reward Level or Reward Type (all $\beta_{mean} < 0.30$, all 95% Bayesian credible interval crossing zero). The full model had lower predictive ability (weight = 0.002) than the model without main and interaction effects of the predictor Drug (weight = 0.998), as shown with LOO-CV. These Bayesian analyses strengthen the view, already conveyed by the frequentist LMMs, that neither drug affected explicit wanting or liking of both types of rewards in this study.

## Implicit measures: facial EMG

To investigate drug effects on reward anticipation and reward consumption, facial EMG was analyzed in relation to trial-by-trial subjective ratings and effort (as continuous predictors) in four periods of interest (see *Figure 1*). In short, the following results were found. In the Pre-Effort anticipation period (*Figure 4*) the corrugator was, as expected, relaxed for greater wanting and effort, and was more activated to food in the amisulpride and naltrexone groups compared to the placebo group. In the same time window, the zygomaticus muscle showed, as expected, stronger activation for greater wanting, however only in food trials. In the Post-Effort anticipation period, a non-significantly greater zygomaticus activation for greater wanting was found. In the Delivery phase, a Liking X Drug interaction was found in the zygomaticus muscle (*Figure 5*), reflecting the expected zygomaticus activation for greater liking in the placebo and (to a lesser extent) amisulpride group, while the opposite pattern of lower zygomaticus contraction for greater liking was found in the naltrexone group. Importantly, this interaction did not survive FDR correction (p=0.09) but seems credible based on a Bayesian LMM. Finally, in the Relax window immediately following reward administration, the corrugator significantly relaxed for the most liked food rewards, but not touch rewards.

## Pre-Effort anticipation

For the corrugator muscle by Wanting, significant main effects of Reward Type ($F(1, 267.9)=10.31$, p=0.01), Wanting ($F(1, 238.5)=7.75$, p=0.01), and Block ($F(1, 7867.4)=7.76$, p=0.01) were found. Activation of the corrugator was greater for food (M = 116.35, SD = 84.72) than touch (M = 110.21, SD = 60.63) and decreased, as expected, with increasing ratings of wanting (slope b = −2.80; *Figure 3A*). A significant Drug X Reward Type interaction ($F(2, 267.8)=4.08$, p=0.04) reflected (*Figure 4D*) greater corrugator activation to food than touch in the amisulpride group (p=0.006; food: M = 119.12, SD = 95.82; touch: M = 109.46, SD = 53.44) and naltrexone group (p=0.001; food: M = 120.00, SD = 101.29; touch: M = 109.66, SD = 67.25), while the placebo group had similar activations across both reward types (p=0.66; food: M = 110.30, SD = 48.29; touch: M = 111.44, SD = 59.87). Corrugator activation to food was also significantly greater in the amisulpride and naltrexone groups compared to the placebo group (p=0.03 and. 01).

For the corrugator muscle by Effort, we found a significant main effect of Reward Type ($F(1, 277.2)=11.04$, p=0.008), with greater activation for food (M = 116.35, SD = 84.72) than touch (M = 110.21, SD = 60.63), a significant main effect of Effort ($F(1, 207.9)=6.38$, p=0.04) due to greater corrugator relaxation with increasing levels of Effort (b = -2.59; *Figure 4B*) a significant main effect of Block ($F(1, 7662.4)=5.72$, p=0.04) due to greater corrugator activation in the second (M = 115.55,

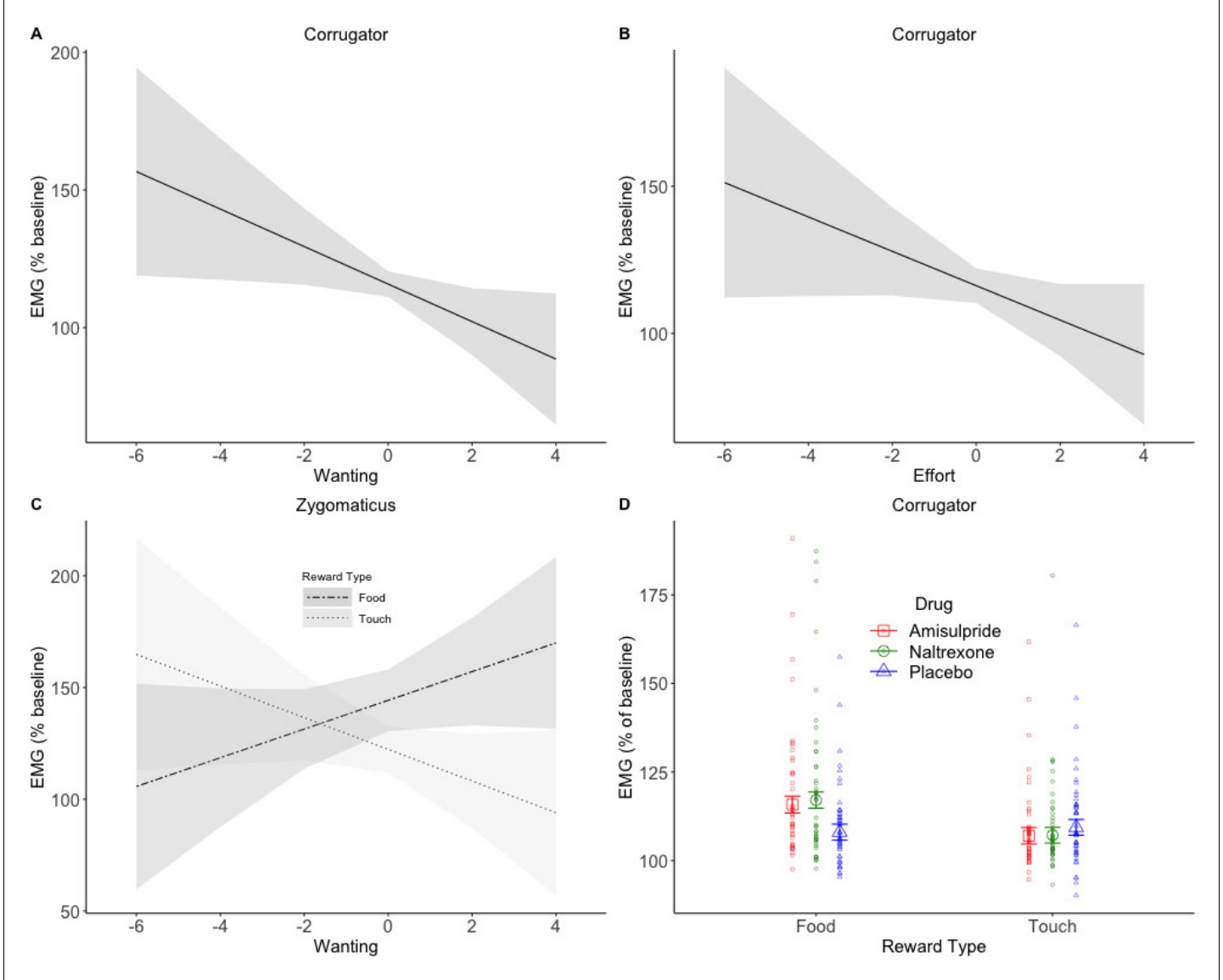

**Figure 4.** EMG during Pre-Effort anticipation. The corrugator relaxed in trials with greater wanting (**A**) and greater effort (**B**). Zygomaticus activation to food (**C**) showed the opposite pattern of activation for trials with greater wanting. This is in line with the literature and was also expected for touch. A significant Drug X Reward Type interaction was found in the corrugator analyses by wanting (**D**), and by effort (not shown). The anticipation of food rewards resulted in greater corrugator activation in the two drug groups compared to placebo, suggesting a reduction in hedonic facial responses. Plots A-C show marginal means and 95% CIs; plot D shows marginal means with standard errors and averages by subject.

*SD* = 87.5) compared to the first block (*M* = 110.98, *SD* = 56.4), and a significant Drug X Reward Type interaction (*F*(2, 277.2)=4.03, p=0.04) reflecting greater corrugator activation for food than touch in the amisulpride group (p=0.007; food: *M* = 119.12, *SD* = 95.82; touch: *M* = 109.46, *SD* = 53.44) and naltrexone group (p<0.001; food: *M* = 120.00, *SD* = 101.29; touch: *M* = 109.66, *SD* = 67.25), while the placebo group had similar activations across both reward types (p=0.72; food: *M* = 110.30, *SD* = 48.29; touch: *M* = 111.44, *SD* = 59.87). Corrugator activation for food was also significantly greater in the naltrexone group compared to the placebo group (p=0.03).

For the zygomaticus muscle by Wanting (random slopes for the Reward Type X Effort interaction were removed to allow model convergence), a significant main effect of Reward Type (*F*(1, 125.9)=13.78, p=0.001) was found, reflecting greater zygomaticus activation for food (*M* = 138.79, *SD* = 145.48) than touch (*M* = 122.98, *SD* = 130.67). Moreover, greater wanting predicted zygomaticus contraction in food trials (b = 5.6) but not in the touch trials (b = −2.73), as shown by a

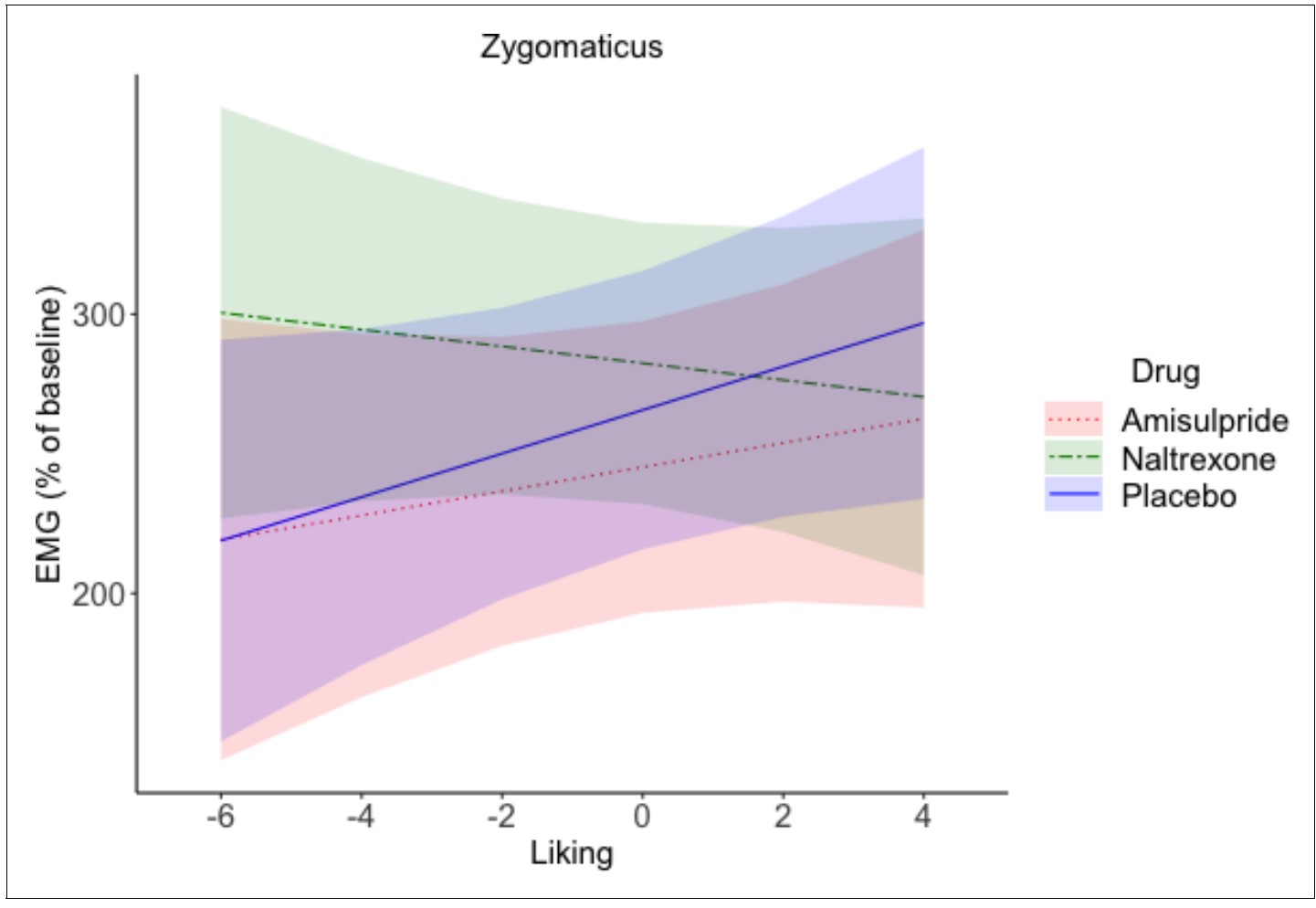

**Figure 5.** Zygomaticus during Delivery (marginal means and 95% CIs). A Liking x Drug interaction (not significant after FDR correction, p=0.09, but credible according to a Bayesian LMM) reflected zygomaticus activation for greater liking in the placebo group, and to a lesser extent also in the amisulpride group, but zygomaticus relaxation in the naltrexone group. This suggests that blocking of the opioidergic system resulted in an inverted effect of liking on zygomaticus activation, with less smiling for the most liked rewards.

significant Reward Type X Wanting interaction ($F$(1, 2925.7)=6.62, p=0.04; *Figure 4C*). All other effects were not significant (all $F < 0.8$, all p>0.68).

For the zygomaticus muscle by Effort (random slopes for the Reward Type X Effort interaction were removed to allow model convergence), only a significant main effect of Reward Type was found ($F$(1, 129.2)=14.70, p=0.001), with greater zygomaticus activation to food ($M = 138.79$, $SD = 145.48$) compared to touch ($M = 122.98$, $SD = 130.67$). All other effects were not significant (all $F < 1.2$, all p>0.81).

## Post-Effort anticipation

No significant effects were found for the corrugator muscle, neither by Wanting nor by Effort (all $F < 1.7$, all p>0.78).

For the Zygomaticus, a greater contraction for increasing levels of Wanting (b = 6.82) was observed, but the effect fell short of significance ($F$(1, 186.9)=6.55, p=0.08). All other effects were not significant (all $F < 1.5$, all p>0.51).

## Reward delivery

Analysis of the corrugator resulted in a significant main effect of Reward Type ($F$(1,121.9) = 8.21, p=0.04), due to greater corrugator activation in response to food ($M$ = 153.25, $SD$ = 216.96) than touch ($M$ = 117.23, $SD$ = 298.61). All other effects were not significant (all $F$ < 2.4, all p>0.48).

For the zygomaticus a significant main effect of Reward Type ($F$(1, 126.1)=77.97, p<0.001), was observed, due to greater zygomaticus activation in response to food ($M$ = 264.83, $SD$ = 233.77) than touch ($M$ = 125.21, $SD$ = 211.39). A Liking X Drug interaction, falling short of significance after FDR correction ($F$(2, 132.2)=4.22, p=0.09, *Figure 5*), was also found. Greater liking resulted, as expected, in greater zygomaticus activation in the placebo group ($b$ = 5.74), and to a lesser extent also in the amisulpride group ($b$ = 0.56). By contrast, the opposite pattern was found in the naltrexone group, with a negative slope ($b$ = −7.86) that was significantly different from the placebo group (p=0.006) but not from the amisulpride group (p=0.09). The amisulpride and placebo group did not differ significantly between each other (p=0.3).

To further probe the Liking X Drug interaction, we also ran a Bayesian LMM with the same predictors (dummy coding was applied to the drug groups amisulpride and naltrexone, to compare them to placebo). A normal, and a half student-t prior were chosen for, respectively, population-level (fixed), and group-level (random) effects. Results confirmed a credible difference, compared to placebo, in the effect of liking on the zygomaticus activation in the naltrexone group ($\beta_{mean}$ = 13.62, 95% Bayesian credible interval [−23.37,−3.68]), but not in the amisulpride group ($\beta_{mean}$ = -4.92, 95% Bayesian credible interval [−14.99, 5.09]).

## Relax phase

For the corrugator by Liking (the random slope for the Reward Type X Liking interaction was removed to allow model convergence), significant main effects of Reward Type ($F$(1, 155.0)=20.36, p<0.001) and Liking ($F$(1, 184.6)=12.41, p=0.001), and a significant Reward Type X Liking ($F$(1, 9231.8)=7.66, p=0.01) interaction were found. The interaction reflected a significant corrugator decrease with greater liking for food rewards ($b$ = −23.2) but not for touch rewards ($b$ = −5.6).

For the zygomaticus by Liking (the random slope for the Reward Type X Liking interaction was removed to allow model convergence), a significant main effect of Reward Type was found ($F$(1, 126.7)=144.25, p<0.001), reflecting greater zygomaticus contraction to food ($M$ = 211.17, $SD$ = 162.63) than touch ($M$ = 132.05, $SD$ = 208.32). Zygomaticus contraction was overall greater in the second ($M$ = 166.22, $SD$ = 148.85) compared to the first block ($M$ = 175.58, $SD$ = 225.67), as indicated by a significant main effect of Block ($F$(1, 9472.0)=7.44, p=0.03).

No significant main or interaction effects for the factor Drug were found in the CS and ZM data (all $F$ < 1.7, all p>0.19).

## Discussion

By adopting a newly developed experimental paradigm (*Korb et al., 2020*), in which reward processing is operationalised similarly to animal research, in combination with a dopaminergic and opioidergic drug challenge, we aimed to address two fundamental and as of yet unresolved research questions: (1) to what extent do motivational and hedonic responses in adult humans rely on shared or separate neurochemical systems, and (2) does the neurochemical basis of human reward processing differ for touch and food rewards.

Analyses of the behavioral (subjective wanting and liking ratings and effort) and physiological data (fEMG during anticipation and consumption of the reward) in relation to drug administration led to the following main results: (1) neither ratings of wanting nor liking were modulated by the pharmacological challenge (as confirmed with Bayesian analyses); (2) participants under dopaminergic or opioidergic antagonists produced significantly lower effort to obtain food rewards (*Figure 3A*), and non-significantly (p=0.056) lower effort to obtain low rewards of both touch and food (*Figure 3B*); (3) during the Pre-Effort anticipation of food, significantly higher corrugator activation was found in both the amisulpride and naltrexone groups (*Figure 4D*), suggesting lower hedonic anticipatory pleasure; and (4) during reward Delivery a Drug X Liking interaction was found (p=0.09 after FDR correction, but confirmed by Bayesian analyses), which reflected greater zygomaticus activation for liked rewards (and thus greater hedonic pleasure) in the placebo and to a lesser

extent in the amisulpride groups, but weaker zygomaticus activation for liked rewards (less hedonic pleasure) in the naltrexone group (*Figure 5*). These findings are now discussed in relation to our main research questions.

## Separate neurochemical systems underlie motivational and hedonic responses

In line with animal models and recent human pharmacological studies, indicating that both the dopaminergic and the opioidergic systems underlie the motivation to obtain rewards (*Chelnokova et al., 2014*; *Peciña and Smith, 2010*; *Weber et al., 2016*), we observed a similar effect of the D2/D3 antagonist amisulpride and the non-specific opioid receptor antagonist naltrexone on the effort produced to obtain the announced reward, resulting in a reduction of applied force. Notably, and differently from our original hypothesis, the effect was most pronounced for the second-preferred (low) rewards, as indicated by a Drug x Reward Level interaction (which however was not significant, p=0.056). A possible explanation for this finding is that our food stimuli did not vary in caloric content (i.e. the three reward levels were matched for fat and sugar). Therefore, individual preferences were derived from other mechanisms than energy value, possibly leading to a different effect of the drug on food reward processing (*Barbano et al., 2009*; *Salamone et al., 2007*). Another possible explanation is that high rewards are less susceptible to changes in their incentive salience when more options are available. Indeed, the majority of the studies in animals and humans have used only two reward levels, and have found that interference with dopaminergic or opioidergic transmission alters the outcome of cost/benefit analyses involving work-related response costs for the most valuable option (*Salamone et al., 2007*). Our finding suggests a similar shift in cost/benefit that is possibly sensitive to a different experimental set-up (*Barbano et al., 2009*).

A similar effect of both amisulpride and naltrexone during food *anticipation* was observed in the implicit measure of fEMG: corrugator activation was greater in both drug groups, compared to placebo. Because frowning typically reflects a more negative (or less positive) reaction to rewards and emotional stimuli (*Fernández-Dols and Russell, 2017*; *Heller et al., 2011*; *Lang et al., 1993*; *Larsen et al., 2003*), the observation that dopamine and opioid antagonists led to greater frowning might be interpreted as a reflection of less anticipated pleasure, independently of reward level, in these groups of participants.

During reward *consumption*, only the opioidergic antagonist naltrexone had an effect on implicit hedonic facial reactions. Lower zygomaticus activation for greater liking was found in the naltrexone group (with both frequentist and Bayesian LMMs), and this effect was significantly different from the placebo group, who instead showed the expected pattern of higher zygomaticus activation for greater liking. The amisulpride group showed the same pattern as the placebo group, although to a lesser extent. We interpret this finding as less smiling to liked rewards after administration of the opioid receptor antagonist naltrexone, as it parallels observations of fewer orofacial hedonic reactions to most preferred foods after opioidergic blockage in animals (*Smith and Berridge, 2007*).

Zygomaticus activation is also known to increase for highly negative stimuli, in addition to positive stimuli (*Lang et al., 1999*; *Larsen et al., 2003*), which could suggest that greater zygomaticus activation with decreasing liking in the naltrexone group is due to disgust expressions, or other negative facial reactions to the least-liked rewards, rather than to a reduction in positive hedonic responses (smiling). This, however, seems unlikely, as we only administered positive rewarding stimuli, and because drug groups did not differ in (1) initial reward preferences (*Figure 2*); (2) the number of high, low, and verylow rewards received; (3) the ratings of wanting and liking of these rewards; (4) the change of ratings of liking over time (*Figure 2—figure supplement 1*); and (5) amounts of nausea or other side-effects (*Table 1*). Moreover, this finding is unlikely to be explained by mouth movements related to food ingestion, because participants were instructed to swallow the food after the Delivery window, and no interaction with the factor Reward Type was found.

Taken together, and partially in line with the behavioral results (where the effects of the drugs were only observable for effort, but not for subjective wanting and/or liking ratings), fEMG data suggest a differential action of dopaminergic and opioidergic drug manipulation during the *anticipation* and *consumption* of rewards, with an effect of both drugs during the anticipation of rewards, but only of naltrexone during subsequent reward consumption. Interestingly, while the corrugator showed a general increase due to drug administration, zygomaticus activity was only affected in its relationship to subjective pleasure, and not in terms of overall activation. This differential pattern

suggests that the two muscles, despite both tracking changes in hedonic value (*Lang et al., 1993*; *Larsen et al., 2003*), do not necessarily behave in a complementary, but rather in an independent way. This might also explain the heterogeneity of findings in previous studies, which often report an effect of reward valence on only one of the two muscles.

Interestingly, drug effects were found on effort levels and fEMG, but not on subjective ratings of wanting and liking. While this may come as a surprise, it is in line with several previous studies, which have reported either null or weak effects of pharmacological interventions on pleasantness likings of affective touch (*Case et al., 2016*; *Ellingsen et al., 2014*; *Løseth et al., 2019*; *Trotter et al., 2016*). However, several studies have reported significant effects of naltrexone/morphine on food liking/ consumption (*Bertino et al., 1991*; *Eikemo et al., 2016*; *Yeomans and Gray, 1996*; *Yeomans and Gray, 1997*; but see *Hetherington et al., 1991* for a null effect). One possibly relevant difference between some of the previous work, and our study, is that we kept calory intake constant across food stimuli, and thus across participants. More research will be needed to clarify if drug-induced changes in reward pleasantness can be reliably assessed with explicit measures (ratings) for some types of rewards (food), but instead require more implicit measures (facial responses, effort) for other types of rewards (affective touch).

### Distinct neurochemical bases for food and touch rewards

Inclusion of both food and touch rewards allowed us to indirectly address the yet unresolved question (*Ruff and Fehr, 2014*), whether different types of rewards are processed by the same neurobiological systems (as proposed by the 'common currency hypothesis'), or if representations coding for different rewards occur in distinct neural circuits, albeit on a common scale (*Grabenhorst and Rolls, 2011*). In particular social rewards, like affective touch, may constitute a separate class of stimuli, with a dedicated neural circuitry (*Rademacher et al., 2010*), which can be specifically impaired, for example in people with autism spectrum disorders (*Chevallier et al., 2012*; *Haggarty et al., 2020*). Although the magnitude of the two types of rewards in terms of subjective ratings and effort was carefully matched (*Korb et al., 2020*), most drug effects were either stronger or restricted to food trials, as indicated by significant Drug x Reward Type interactions for measures of effort to obtain the announced reward, and for corrugator activation during Pre-Effort *anticipation*. This suggests that the decision utility of touch and food rewards may not rely on the same neurochemical brain systems. However, fEMG responses to food were also stronger to begin with, as indicated by significant main effects of Reward Type for both muscles during the Pre-effort anticipation, Delivery, and Relax analysis windows. This might explain why only reactions to food rewards were modulated by opioidergic and dopaminergic antagonists. Another possible explanation for the less pronounced drug effects for touch is that responses to social rewards, including touch, might also depend on oxytocin and serotonin, in addition to dopamine and opioids (*Fischer and Ullsperger, 2017*; *Tang et al., 2020*; *Walker and McGlone, 2013*). This is also suggested by the finding of higher pleasantness ratings and greater zygomaticus activation to touch after administration of 3,4-methylenedioxymethamphetamine (MDMA), a drug that modulates serotonin, dopamine, and possibly oxytocin levels (*Bershad et al., 2019*; *de Wit and Bershad, 2020*).

Of note, drug effects on the activity of the zygomaticus muscle during reward *consumption* were similar for touch and food rewards, as indicated by the absence of a Drug X Reward Type interaction during the Delivery and Relax periods. To the best of our knowledge, this is the second study (after *Bershad et al., 2019*) to report a pharmacological modulation of hedonic responses to experienced touch (for a weaker effect see also *Case et al., 2016*). Major differences in our study compared to previous work (*Ellingsen et al., 2014*; *Løseth et al., 2019*) are the delivery of affective touch with the hand instead of a brush (*Ellingsen et al., 2014* included touch by hand but wearing a glove), and allowing participants to select their preferred touch speed. Regarding the way touch was delivered, it is possible that the social saliency of the touch stimuli delivered through direct skin contact was enhanced, compared to when the touch is delivered with a brush, allowing us to detect subtle effects of the drug. Regarding the selection of the preferred touch speed, even if the majority of our subjects selected the slower speed as the preferred one, inter-individual differences were observed, and the implementation of a task that could account for those, may have helped to detect the effect of the drug. Further studies should investigate how such factors modulate drug responses to perceived affective touch.

The current study is characterized by a number of limitations. First, only two types of stimuli (food and touch) were used to define social and non-social rewards, and only two neurochemical systems (dopaminergic and opioidergic) were challenged. However, from the animal and human literature, we know that other systems (e.g. endocannabinoids, orexin, benzodiazepine, etc.) also contribute to the motivational and hedonic components of reward processing (*Berridge and Kringelbach, 2015*). Future studies should therefore broaden the neuropharmacological investigation of social vs. nonsocial reward processing, by using other compounds and different rewarding stimuli, to allow for the generalizability of the findings to other social and nonsocial rewards, and to better understand their neurochemical basis. Furthermore, computational approaches will most likely be useful to reveal hidden psychological states subtending motivation and experienced pleasure, allowing to refine how drug administration acts on these two components (*Meyer et al., 2019*).

Second, we used a relatively low dose of amisulpride (400 mg). Amisulpride can both increase dopaminergic neurotransmission by blocking presynaptic autoreceptors when given at low doses (50–300 mg), and decrease it by blocking postsynaptic D2/D3 receptors when given at higher doses of 400–1200 mg (*Racagni et al., 2004*; *Schoemaker et al., 1997*). The used dose of 400 mg is at the lower end of postsynaptically active high doses (*Rosenzweig et al., 2002*) and occupies ~70% of D2 receptors when given for 2 weeks (*Meisenzahl et al., 2008*). We decided on 400 mg of amisulpride based on previous studies in humans (e.g. *Weber et al., 2016*), to obtain postsynaptic dopaminergic effects, while ensuring the safety and well-being of our participants, as well as allowing both participants and experimenters to remain in the dark about the type of compound or placebo administered (double blinding). The effects we observed (e.g. less effort) are in line with amisulpride's antagonistic action on postsynaptic D2/D3 receptors. Upon availability of drug compounds that more strongly modulate the dopaminergic systems with minimal side effects, future studies should, however, explore dose-dependent changes in human reward processing.

Third, we used a cross-sectional design for drug/placebo administration. A within-subjects design would certainly have resulted in greater statistical power. However, this would have come with the cost of even greater habituation to the rewards.

Fourth, the study suffered from a lack of power to detect small effects. We had modeled the sample size on a previous study using the same drugs and doses (*Weber et al., 2016*). However, *Weber et al., 2016* only found relatively small drug effects, and several other studies have failed to show effects of a pharmacological modulation of the opioid, serotonin, or oxytocin systems on the liking of affective touch (*Ellingsen et al., 2014*; *Løseth et al., 2019*; *Trotter et al., 2016*). This reveals the difficulty of uncovering the neurochemical basis of reward processing in humans and suggests that larger sample sizes should be used in future pharmacological studies to investigate the neurochemical bases of touch and other rewards.

## Conclusion

We report pharmacological evidence in healthy human volunteers, across several measures including the monitoring of facial expressions with fEMG, about the role of the dopaminergic system for the motivational component and of the opioidergic system for both motivational and hedonic components of reward processing. The effort to obtain a reward and valenced facial reactions during reward anticipation were both modulated by the administration of dopaminergic or opioidergic antagonists. By contrast, facial reactions during reward experience were only altered by the opioidergic antagonist, suggesting neurochemical differences underlying hedonic expressions during *anticipation* and *experience* of pleasure. Explicit ratings of reward wanting and liking were not modulated by either drug. This constitutes the first demonstration of this kind in adult humans, using an operationalization of reward closely resembling previous animal research, and it suggests that the neurochemical regulation of pleasure (as indicated by hedonic facial reactions) is phase-specific, depending on whether the reward is anticipated or experienced. The finding that most drug effects were either stronger for, or restricted to, food trials may indicate different neurochemical brain mechanisms for social and nonsocial rewards. This point however requires further investigation via brain imaging or more direct measures of brain activity in addition to pharmacological challenges tailored to investigate the role of different neurochemical systems in the processing of social versus nonsocial rewards.

# Materials and methods

## Subjects

Based on previous work that had used the same compounds and doses (*Weber et al., 2016*), we aimed at collecting data from 40 participants per group or more. The final study sample included 131 volunteers (88 females) aged 18–35 years (*M* = 23.3; *SD* = 3.5). In the amisulpride group, blood concentrations of the drug (measured 5 hr after intake) were in or above the therapeutic range (blood samples missing for six people). Specifically, the minimum was 212 ng/mL, and 19 participants were above 604 ng/mL. All participants reported being right-handed, to smoke less than five cigarettes daily, to have no history of current or former drug abuse, to like milk and chocolate, not to suffer from diabetes, lactose intolerance, lesions or skin disease on the left forearm, and to be free of psychiatric or neurological disorders. Participants' average Body Mass Index (BMI) was 22.6 (*SD* = 2.5, range 17.7–29.3). To reduce the chances that social touch would be perceived as a sexual reward, the touch stimulation was always carried out by a same-sex experimenter (see Procedure), and only participants who reported to be heterosexual were included. The study was approved by the Ethical Committee of the Medical University of Vienna (EK N. 1393/2017) and was performed in line with the Declaration of Helsinki (*World Medical Association, 2013*). Participants signed informed consent and received monetary compensation of 90€.

## Stimuli

Three stimuli with identical fat and sugar content (1.5 g fat, 10 g of sugar per 100 g) were used as food rewards: milk, chocolate milk, and a 4:1 mix of milk and chocolate milk. Tap water served for rinsing at the end of each trial. The initial stimulus temperature of these liquids was kept constant (~4°C) across participants. Stimulus delivery was accomplished through computer-controlled pumps (PHD Ultra pumps, Harvard Apparatus) attached to plastic tubes (internal ø 1,6 mm; external ø 3,2 mm; Tygon tubing, U.S. Plastic Corp.), which ended jointly on an adjustable mount positioned about 2 cm in front of the participant's mouth. In each trial, 2 mL of liquid was administered for 2 s. Overall, including stimulus pretesting (see Procedure), participants consumed 196 mL of liquids, composed of 98 mL of water, and 98 mL of sweet milk with different concentrations of chocolate aroma (depending on effort, see below).

Touch rewards consisted of gentle caresses over a previously-marked 9-cm area of the participant's forearm (measurement started from the wrist towards the elbow). Three different caressing frequencies, chosen based on the literature and pilot testing, were applied for 6 s by a same-sex experimenter: 6 cm/s, 21 cm/s, and 27 cm/s. To facilitate stroking, the stimulating experimenter received extensive training and, in each trial, heard rhythmic sounds, indicating the rhythm for stimulation, through headphones.

## EMG

After cleansing of the corresponding face areas with alcohol, water, and an abrasive paste, reusable Ag/AgCl electrodes with 4 mm inner and 8 mm outer diameter were attached bipolarly according to guidelines (*Fridlund and Cacioppo, 1986*) on the left corrugator supercilii (corrugator) and the zygomaticus major (zygomaticus) muscles. A ground electrode was attached to the participants' forehead and a reference electrode on the left mastoid. The EMG data were sampled at 1200 Hz with impedances below 20 kOHM using a g.USBamp amplifier (g.tec Medical Engineering GmbH) and the software Matlab (MathWorks, Inc).

## Procedure

A monocentric, randomized, double-blind, placebo-controlled, three-armed study design was used. The study took place in the Department of Psychiatry and Psychotherapy at the Medical University of Vienna. Participants visited the laboratory for the first visit (T0) in which they received a health screening, followed by a second visit (T1) that included oral drug intake and the experiment described here. Pharmacological dosage, and length of waiting time after drug intake (3 hr), were modeled on previous work (*Weber et al., 2016*), and on the drug's pharmacodynamics. Amisulpride reaches the first peak in serum after 1 hr, and a second (higher) peak after approximately 4 hr. The elimination half-life is 12 hr (*Rosenzweig et al., 2002*). At doses of 400 mg or higher, amisulpride

acts as a postsynaptic D2/D3 receptor antagonist and thus results in lower dopaminergic action (*Racagni et al., 2004*; *Schoemaker et al., 1997*). Naltrexone reaches maximal concentration in plasma after 1 hr, has an elimination half-life in plasma of approximately 4 hr, and is completely cleared from plasma after 96 hr (*Meyer et al., 2019*). Importantly, up to 90% of mu-opioid receptors in the brain remain blocked by naltrexone after 48 hr, and partial receptor blockade could be shown up to 168 hr after intake (*Lee et al., 1988*).

Participants came to T1 with an empty stomach (it was morning, and they had been instructed not to eat in the preceding 6 hr), filled out the PANAS questionnaire, tested negative (or were excluded) on a urine drug screen sensitive to opiates, amphetamine, methamphetamine, cocaine (among other things), and then received a capsule filled with either 400 mg of amisulpride (Solian), 50 mg of naltrexone (Dependex), or 650 mg of mannitol (sugar) from the study doctor. All capsules looked identical from the outside, and neither participants nor the experimenters were informed of their content. Drug intake was followed by a waiting period, EMG preparation, and task instructions.

The experiment comprised two tasks following procedures described elsewhere (*Korb et al., 2020*). The main task started 3 hr after pill intake. Participants were seated at a table and comfortably rested their left forearm on a pillow. A curtain blocked their view of the left forearm and the rest of the room. This was particularly relevant for touch trials, in which one of two same-sex experimenters applied the touch rewards to the participant's left forearm. Two experimenters were always present during testing, to limit the influence of participants' experimenter preferences, and to allow participants to better concentrate on the (touch) stimuli.

Participants first completed a short task, in which they experienced and individually ranked three food rewards, and separately three touch rewards, presented randomly in sets of three of the same reward type. In the main task, which started 3 hr after pill intake, the previously most liked stimuli were used as 'high' rewards, the stimuli with medium liking as 'low' rewards, and the least liked stimuli were used as 'verylow' rewards. To calibrate the dynamometer, the MVC was established right before the short task, by asking participants to squeeze the dynamometer (HD-BTA, Vernier Software and Technology, USA) with their right hand as hard as possible three times, each lasting 3 s . The average MVC (peak force in newtons across all three trials) was 212 ($SD$ = 80.4) and did not differ significantly between drug groups, as tested by linear regression ($\beta$ = 1.6, $SE$ = 8.68, $t$ = 0.19, p=0.85).

After calibration of the dynamometer, EMG electrodes were attached, participants received detailed instructions, and completed four practice trials (two per reward type). The main task included four experimental blocks with 20 trials each. Each block contained either food or touch trials, and the blocks were interleaved (ABAB or BABA) in a counterbalanced order across participants. Each trial included the following steps (*Figure 1*; see *Figure 1—figure supplement 1* and Supporting Information for all elements of a trial): (1) a picture announcing the highest possible reward (high or low, 3 s), (2) a continuous scale ranging from 'not at all' to 'very much' to rate (without time limit) wanting of the announced reward (ratings were converted to a Likert scale ranging from −10 to +10), (3) a 4 s period of physical effort, during which probability of receiving the announced reward was determined by the amount of force exerted by squeezing the dynamometer with the right hand, while receiving visual feedback (sliding average of 1 s, as percentage of the MVC), (4) a picture announcing the obtained reward (3 s for food, 7.3 s for touch), which could be high, low, or – if insufficient effort had been exerted – verylow (the greater participants' effort, the higher the probability of obtaining the announced reward), (5) a phase of reward delivery (2 s for food, 6.5 s for touch – this difference in timing was necessary to obtain sufficiently long tactile stimulation, while keeping the overall trial duration similar across reward types), (6) for food trials instructions to lean back and swallow the obtained reward (duration 3 s), (7) a relaxation phase (5 s), and (8) a continuous scale to rate the liking of the obtained reward. In food trials, participants then received water for mouth rinsing. In both reward types, trials ended with a blank screen for 3–4 s. The last four trials in each block did not require pressing of the dynamometer. These trials were added to the design, in case participants would never press at all, which did not happen for any participant. Trials without pressing were kept in the data, as removing them from analyses reduced power but did not change the pattern of results. After each block participants were allowed to take a short break.

Both tasks were run on a desktop computer with Windows seven using MATLAB 2014b and the Cogent 2000 and Cogent Graphics toolboxes and presented on an LCD monitor with a resolution of 1280 × 1024 pixels. The positive and negative affect schedule (PANAS; *Watson et al., 1988*), and a

questionnaire assessing nausea and 50 other side effects, was filled out twice at the main laboratory visit: just before pill intake, and 3 hr later. Levels of amisulpride (ng/mL) were measured in blood samples taken 5 hr after pill intake (after both tasks).

## Analyses

Data and analysis scripts are available online (https://osf.io/vu8dz). Group comparisons for age, BMI, MVC, PANAS scores, and side effects, were made with linear regressions using the lm() function. Differences in the ranking of rewards across drug groups were tested with separate ordinal regressions by Reward Type (food, touch), using the package *ordinal*.

All other analyses were done with linear mixed-effects models (LMMs), fitted through restricted maximum likelihood (REML) estimation, using the lmer() function of the *lmerTest* package in R (which adds p values to the lme4 output; *Bates and Maechler, 2014*; *R Development Core Team, 2019*), and with Helmert contrast coding. In comparison to ANOVAs, LMMs reduce Type-I errors and allow for a better generalization of findings (*Judd et al., 2012*). To control for the effect of time – possibly inducing fatigue and/or habituation (*Figure 2—figure supplement 1*) – the four blocks were recoded to two blocks by Reward Type and entered as covariates to the LMMs. Figures (except *Figure 1*) were created in R using the packages *ggplot2, ggpirate, and cowplot*.

Behavioral data were analyzed in the following manner. Outlier trials were defined as those with a rating of wanting, rating of liking, or amount of exerted force, which was greater/smaller than the subject's mean +/- 2 times the subject's standard deviation. This led to an average rejection of 6.56 trials per participant (SD = 3.71). The total number of excluded trials did not differ significantly between groups ($t(133) = -1.28$, p=0.20). For each behavioral dependent variable (ratings of wanting and liking, and effort), a LMM was fitted with the fixed effects Reward Type (food, touch), Reward Level (high, low, verylow), Drug (amisulpride, naltrexone, placebo), their interactions, and with Block (first, second) as a covariate. Categorical predictors were centered through effect coding, and by-subject random intercepts and slopes for all within-subjects factors and their interactions were included as random effects (unless the model did not converge, in which case the random-effects structure was gradually simplified, e.g. by first dropping the interaction among within-subjects factors). Type-III F-tests were computed with the Satterthwaite degrees of freedom approximation. We report all statistically significant (p<0.05) effects, and non-significant effects with p<0.1 that are of interest because related to the main hypotheses, as Anova() outputs. Model tables showing all fixed and random effects can be found in the Supporting Information.

Due to technical failure, one participant lacked EMG data entirely, and another participant lacked the EMG for half of the trials. The EMG data were pre-processed in Matlab R2018a (www.themathworks.com), partly using the EEGLAB toolbox (*Delorme and Makeig, 2004*). A 20 to 400 Hz band-pass filter was applied, then data were rectified and smoothed with a 40 Hz low-pass filter. Epochs were extracted focusing on periods of reward *anticipation* (Pre-Effort and Post-Effort anticipation) and reward *consumption* (Delivery and subsequent Relax). EMG was averaged over time-windows of one second, with exception of the 6.5-seconds-long period of touch Delivery, which was averaged over five windows of 1.3 s each, to obtain the same number of windows as for food delivery. We excluded for each participant trials on which the average amplitude in the baseline period (1 s during fixation) of the corrugator or zygomaticus muscles was lower than M−2*SD, or higher than M+2*SD (M = average amplitude over all trials' baselines for the respective muscle and participant). On average, this led to the rejection of 7.7% of trials per participant (*SD* = 2.5). EMG analyses were carried out in four periods of interest: *Pre-effort anticipation* during reward announcement at the beginning of each trial (3 s), *Post-effort anticipation* during the announcement of the gained reward (3 s), *Delivery* (5 s for food and 6.5 s for touch, both averaged to five 1 s time windows), and *Relax* (5 s). For each trial, values in these epochs were expressed as percentage of the average amplitude during the fixation cross at the beginning of that trial. For the Pre- and Post-Effort anticipation periods, separate LMMs were fitted by muscle, with the fixed effects Drug (amisulpride, naltrexone, placebo), Reward Type (food, touch), and either trial-by-trial Wanting or Effort (these were continuous predictors), and all interactions. During the Post-Effort anticipation period, participants could receive the information that they were going to obtain the verylow reward, to which the preceding ratings of wanting and effort did not apply. Because this may have been frustrating for participants, we also carried out analyses excluding trials, in which verylow rewards were obtained. As the results did not change, we kept all trials. For the Delivery and Relax periods, separate LMMs on all trials were fitted

by muscle, with the fixed effects Drug, Reward Type, and Liking. In all LMMs Wanting, Effort, and Liking were centered and scaled by subject, and Block (first, second) was added as a covariate to control for the effects of fatigue or habituation. We controlled for the false discovery rate (FDR) associated with multiple testing of the EMG data using the Benjamini-Hochberg method (*Benjamini and Hochberg, 1995*). Model tables with un-corrected *p*-values can be found in the Supporting Information.

## Acknowledgements

The study was supported by the Vienna Science and Technology Fund (WWTF) with a grant (CS15-003) awarded to Giorgia Silani and Christoph Eisenegger, a grant (VRG13-007) awarded to Christoph Eisenegger, and the Open Access Publishing Fund of the University of Vienna. Funders had no role in study design, data collection and analysis, decision to publish or preparation of the manuscript. We thank Prof. Mathias Pessiglione and his group for help with coding the dynamometer output in Matlab, and Prof. Boris Quednow for commenting on an earlier version of the manuscript. We thank Lukas Lengersdorff, Annabel Losecaat Vermeer, Nace Mikus, Lei Zhang, and Matteo Lisi for providing statistical advice. Many thanks to Mumna Al Banchaabouchi and the interns and master students whose help was crucial for data acquisition: Mani Erfanian Abdoust, Anne Franziska Braun, Raimund Bühler, Lena Drost, Manuel Czornik, Lisa Hollerith, Berit Hansen, Luise Huybrechts, Merit Pruin, Vera Ritter, Frederic Schwetz, Conrad Seewald, Carolin Waleew, Luca Wiltgen, Stephan Zillmer. We are extremely grateful to the reviewers Siri Leknes, Guillome Sescousse, Diego Pizzagalli, and Yuen-Siang Ang, whose comments have helped us to improve the manuscript.

## Additional information

### Funding

| Funder | Grant reference number | Author |
| --- | --- | --- |
| Vienna Science and Technology Fund | CS15-003 | Christoph Eisenegger Giorgia Silani |
| Vienna Science and Technology Fund | VRG13-007 | Christoph Eisenegger |
| University of Vienna | Open Access Publishing Fund | Giorgia Silani |

The funders had no role in study design, data collection and interpretation, or the decision to submit the work for publication.

### Author contributions

Sebastian Korb, Conceptualization, Data curation, Software, Formal analysis, Supervision, Investigation, Visualization, Methodology, Writing - original draft, Project administration, Writing - review and editing; Sebastian J Götzendorfer, Claudia Massaccesi, Investigation, Project administration, Writing - review and editing; Patrick Sezen, Investigation, Medical testing and drug administration; Irene Graf, Investigation, Medical exams and drug administration, curation of analysis of blood samples; Matthäus Willeit, Conceptualization, Project administration, Writing - review and editing; Christoph Eisenegger, Conceptualization, Funding acquisition; Giorgia Silani, Conceptualization, Supervision, Funding acquisition, Project administration, Writing - review and editing

### Author ORCIDs

Sebastian Korb https://orcid.org/0000-0002-3517-3783
Claudia Massaccesi http://orcid.org/0000-0003-0519-6324
Giorgia Silani https://orcid.org/0000-0002-4284-3618

## Ethics

Human subjects: The study was approved by the Ethical Committee of the Medical University of Vienna (EK N. 1393/2017) and was performed in line with the Declaration of Helsinki (World Medical Association, 2013). Participants signed informed consent.

## Decision letter and Author response

Decision letter https://doi.org/10.7554/eLife.55797.sa1
Author response https://doi.org/10.7554/eLife.55797.sa2

# Additional files

## Supplementary files

- Supplementary file 1. Description of all elements in a trial.

- Transparent reporting form

## Data availability

All the data (single trials in long format) and scripts to perform statistical analyses and graphical representations (to be run in the software R) are available on open science framework (https://osf.io/vu8dz). The data are available as tab-delimited. txt files, which can be opened in Excel or other programs. Most variable names are self-explanatory, others are indicated in the scripts.

The following dataset was generated:

| Author(s) | Year | Dataset title | Dataset URL | Database and Identifier |
|---|---|---|---|---|
| Korb S, Gotzendo-fer S, Massaccesi C, Sezen P, Graf I, Eisenegger WMC, Silani G | 2020 | Dopaminergic and opioidergic regulation of implicit hedonic facial reactions during anticipation and consumption of social and nonsocial rewards | https://osf.io/vu8dz | OSF, vu8dz |

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
