## [Decision Letter]

**Acceptance summary:**

This study sheds light into the neurochemical basis of human reward processing by providing a translation approach, inspired by three decades of animal studies, to study the neurochemical basis of wanting and liking rewards in humans. Of interest, this study shows that physical reactions that occur when anticipating or consuming a reward rely on dopamine and/or opioid systems. Further, these studies suggest that the type of reward, social or non-social, can involve different neurochemical systems.

**Decision letter after peer review:**

Thank you for submitting your article "Role of dopamine and opioids on facial reactions in anticipation and consumption of social and nonsocial rewards" for consideration by *eLife*. Your article has been reviewed by three peer reviewers, and the evaluation has been overseen by a Reviewing Editor and Christian Büchel as the Senior Editor The following individuals involved in review of your submission have agreed to reveal their identity: Siri Leknes (Reviewer #1); Guillome Sescousse (Reviewer #2).

The reviewers have discussed the reviews with one another and the Reviewing Editor has drafted this decision to help you prepare a revised submission.

Summary:

This study translates key findings on the dopamine and opioid neurochemical dissociation of liking and wanting in rodents, into healthy humans. Using a well-controlled protocol closely inspired from animal literature the authors show that dopamine receptor and opioid receptor antagonists impact subjective ratings and some measures of facial EMG activity during reward anticipation, whereas only a opioid receptor antagonist affected facial EMG activity during reward consumption. Overall this study provides a significant contribution to the understanding of the neurochemical modulation underlying different aspects of behavioral reinforcement in humans.

Revisions:

The reviewers conclude that the study is of interest for *eLife* and that additional experiments are not needed. However, the reviewers request substantial revisions focused on the analysis and interpretations of the study.

1) The authors modeled their sample size on a previous study that showed small drug effects (Weber et al., 2016). Please clearly acknowledge the lack of power to detect small effects and provide interpretations of the results relevant to these small effects.

2) Please clarify why the main tests for the relationship between implicit and explicit responses were conducted in the full sample, rather than first being demonstrated only in the placebo group. Additionally, please clarify why the results are presented as pooled means instead of showing the mean and variance data for each drug, reward type and reward level. On the basis of the interaction effects with p's lower than 0.05, it is concluded that "low rewards" are affected by opioid and dopamine D2/D3 antagonists. Please demonstrate that this result is indeed comparable for touch and taste rewards.

3) Please clarify if side effects of the drugs were measured? For instance, nausea could affect responses to food stimuli.

4) Outlier removal is commonly applied with psychophysiological data, but not for VAS ratings. Please clarify why the outlier values were removed for these measures in the present study. Additionally, do the results differ with and without the excluded datapoints?

5) Please provide clarity into the details of the task and the outcomes:

a)It appears that the 3 reward levels for food reward were milk, chocolate milk, and a 4:1 mix of milk and chocolate milk? Was the ranking of these 3 reward levels consistent across participants? It is plausible that some participants would prefer chocolate milk while others would prefer milk, and that the strength of this preference (i.e. how far apart liking ratings are) would widely vary between participants. It would be informative to report (in supplementary material) the pre-task liking ratings for the 3 reward levels, both for the food and touch rewards. In particular, it would be nice to show that these pre-task ratings did not differ between groups, as this would strengthen the interpretation of the Drug x Reward type interaction reported in Figure 2C as a genuine effect of amisulpride, rather than a mere effect of pre-existing differences between groups.

b) Expression "if squeezing was not sufficient" and "insufficient effort" might be misleading as it implies that a minimum amount of squeezing (i.e. a threshold) was required to obtain a reward. Please clarify that there is a linear (i.e. continuous) relationship between squeezing and the probability of gaining a reward.

c) Please clarify it there were as many high reward as low reward cues.

6) Based on previous literature, when is the typical peak of maximal efficiency for amisulpride and naltrexone, and does this peak coincide with the 3h delay used in the current study?

7) There are 5 dependent variables (effort, wanting and liking ratings, EMG of CS and ZM muscles) tested at 4 different times (Pre-effort anticipation, Post-effort anticipation, Delivery and Relax), as a function of 3 main conditions (Drug, Reward type, and Reward level). As a result, there is a large number of statistical tests being performed, thus a correction for multiple comparisons would be warranted (see e.g. Cramer et al., Psychon Bull Rev 2016).

8) Due to the repetitive nature of the ratings on every trial, while there's no incentive for the participants to provide faithful ratings, the current procedure may run the risk of producing stereotyped ratings with little variability. Could the authors demonstrate that this is not the case? Further, there are possible satiety effects (i.e. ratings going down gradually, especially for the food reward). Please clarify if this occurred.

9) The authors have demonstrated a dissociation between drug effects on wanting and liking measures. However, Point 2 (“does the neurochemical basis of human reward processing differ for social and nonsocial rewards”) is overstated. The results from one social and one nonsocial stimulus type is insufficient to allow generalization to a class of rewards. Moreover, several drug studies have reported being unable to alter liking and/or wanting measures of stroking touch. This includes morphine, naltrexone, tryptophan and oxytocin. To the reviewers' knowledge, the only drug that alters perception of stroking touch is MDMA. Please provide discussion of the current understanding of the difference between drug effects on food and touch reward.

10) The decreased effort levels observed for food rewards in the amisulpride and naltrexone groups compared to the placebo group convincingly argue in favor of a role of the dopamine and opioid systems in motivation. However, and in contrast to the hypothesis stated in the Introduction, the same pattern wasn't observed for touch rewards. This discrepancy limits the generalizability of the conclusion, and warrants discussion. This also applies to the effect of amisulpride and naltrexone on CS muscle contraction (Figure 4): the fact that these effects are restricted to food reward limits the generalizability of the conclusion that the dopamine and opioid systems are involved in anticipated pleasure.

11) Additionally, for Figure 4, please clarify why reward magnitude was not included as a fixed factor similar to other analyses? Might dopamine and opioid antagonism be dissociable effects in terms of their effect on reward magnitude? Similarly, for effort analysis.

12) In the Discussion the statement "the results speak for a genuine alteration of the incentive salience of the low reward, whose rewarding value approached that of the verylow reward, due to the pharmacological manipulation" might be overstated and not supported by the findings. Subjects were never asked to exert effort to pursue verylow reward, hence, the rewarding/motivational value of verylow reward is unknown and it is inaccurate to conclude that the low reward's "rewarding value approached that for verylow reward". Moreover, Figure 2C showed that subjects liked the low rewards much more than verylow rewards; although this is not motivational salience, it might be reasonable to expect them to be associated to some degree. An elegant way to extract these motivational values would be to apply computational modelling, which has been conducted in other similar effort-based decision-making tasks.

13) In the Discussion the findings do not appear to "corroborate the hypothesis that the dopaminergic system underlies the motivational but not the hedonic component of rewards, while the opioidergic system underlies both." Figure 2C shows that the liking result for placebo is similar to naltrexone and not amisulpride. Using placebo as the reference for comparison, the findings appear to show instead that dopamine antagonism has an influence on increasing hedonism, but opioid antagonism has no effect, which is surprising in the context of existing literature.

14) In the Discussion please clearly state how the results align or do not align with the hypotheses. For instance, the discussion of the naltrexone group's zygomaticus “liking” responses gives the appearance as if the result was expected. However, it is surprising that opioid blockade (1) did not reduce subjective liking for any stimuli, not even high-calorie foods, and (2) appears to have reduced not zygomaticus activity in general, but more specifically its association with ratings of liking.

15) Please provide in the Discussion an interpretation of the effects of naltrexone and amisulpride at the doses chosen.

16) The authors suggest that the CS and ZG measures are complementary, but perhaps they are actually more different than similar? CS assesses frowning, which indicates negativity, whereas ZG measures smiling, which indicates positivity. CS is a proxy for negative valence axis, while ZG measures only positive coordinates on the valence axis; neither is a continuous assessment across the whole spectrum from the negative to positive end. This has implications for the interpretations in the study. For example, the authors state that "greater frowning during reward anticipation might be interpreted as a reflection of less anticipated pleasure", but this could be more appropriately interpreted as greater frustration, which is different from positive anticipation (see e.g. Bremhorst et al., 2019, PMID: 31848389). To further this point, if CS and ZG are indeed complementary measures, one would also expect ZG effect during the Pre-Effort anticipation of food – but that is not seen. Similarly, for reward delivery, while there is a CS effect, there is no ZG effect. What is the association between the two measures? A more in-depth discussion would help to improve the interpretations of the study.

17) How do the authors interpret the finding that during reward delivery, the ZM-liking slope for amisulpride did not differ from placebo nor from naltrexone? Comparisons between naloxone and placebo were interpreted to mean opioid antagonism reduced smiling, but no discussion was made for dopamine antagonism.

18) It is unclear from the main text or Materials and methods what feature of the force-time curve is used as the proxy of effort exerted – later in the legend of Figure 3—figure supplement 1, it is stated that "exerted force is the maximum value reached in the 4-sec period". If that is correct, please state this more clearly within the manuscript and also explain, what is the rationale for adopting the peak in a single infinitestimal timepoint within the 4-sec window as the measure of exerted force? Based on experience with hand-dynamometers in physical effort paradigms, it is not easy to control the force one intends to exert. Hence, there is typically an initial sharp spike in the force-time curve where the peak occurs (but not necessarily reflecting what the subject intends to exert), before stabilizing to a plateau that more accurately reflects the subjects intended force exerted. Thus, it is more typical in other studies to require subjects to sustain their force for say, 2s out of a 4-s window, and then take that sustained force level as a proxy of effort exerted.

19) Further, how is force exerted converted to probability? What constitutes as insufficient effort? The probability component seem an unnecessary manipulation thus what was the authors' rationale for rewarding probabilistically, rather than delivering the reward as long as sufficient effort is exerted?

20) A substantial number of subjects have a mean force exertion much higher than 100% of their MVC, especially in the amisulpride group (Figure 3—figure supplement 1) – does that mean the baseline measurement of MVC is not accurate since it is not actually maximum? Alternatively, it might also reflect issues associated with taking peak at the spike of the force-time curve as the measure of force exerted.

21) The authors do not account for the effect of fatigue in their analyses, which is presumably substantial given the large number of trials they have to squeeze the hand-grip dynamometer. The authors might include a factor of block?

22) Please provide rationale behind the choice of variables in the LMM, which includes all fixed factors and their interactions, as well as subject random intercepts and random slopes for all within-subject factors and their interactions. This appears as overparameterization. It is not easy to detect overparameterization since each random effect uses only one df, but the estimation of variance among small numbers of groups can be numerically unstable. The approach of fitting a model as complex as possible is problematic as it makes it very easy to fit a model too complex for the data. If the model fails to converge, that is fine since inference is not possible. But if the model "works", then that risks false inference (see e.g. Zurr et al., 2010).

23) The authors considered only one possible model. From a statistical point of view, however, the full range of possible interactions, from the full model (including all possible interactions between factors) to just the main effects, should be considered and the best model analyzed according to e.g. AIC/BIC.

24) Please provide more details on how the LMM analyses were conducted. For example, did they use ML or REML, and why?

25) Please provide tables denoting all the parameter estimates, SE, F-statistic, p-value etc. for each LMM.

26) In Results paragraph one: There are 80 trials in total across four blocks, but the total number of rewards obtained do not seem to add up to close to 80? Please clarify that the mean rewards reported are collapsed across condition.

Additional major comments to improve the clarity of the study:

27) The Introduction is very long. A more concise Introduction would improve the overall clarity. Further, please address the following in the Introduction:

a) Wanting and liking have been linked primarily to dopamine and opioids, but it's useful to extend this rule of thumb to encompass evidence on endocannabinoids, orexin and even benzodiazepines in enhancing liking responses to a comparable degree as mu, kappa and delta opioid microstimulation. This perspective should also be considered in the conclusion section which should be rephrased to avoid the impression that you have tested every relevant neurotransmitter.

b) Please provide an explanation for the interpretation that opioid-induced increases in motivation for high-calorie foods as evidence that opioids primarily affect wanting through liking.

c) Please consider changing to past tense when reviewing the evidence from single/small numbers of studies. Also, the effects reported in Chelnokova et al., 2014 on wanting were not explicitly subjective, rather a count of button presses which prolonged or shortened the duration an image was shown on the screen. Other relevant references on opioids and liking/wanting in humans include Gospic et al., 2008, Psychopharmacol; Atlas et al., 2013; Eikemo et al., 2017; Chelnokova et al., 2016, SCAN. Importantly, Gospic, Atlas and colleagues tested both men and women, whereas the cited evidence from Leknes' lab and Büchel's erotic image study were conducted exclusively in men. (Note that the affective image stimuli used by Gospic, Atlas and colleagues could also be described as social, and studies from Harriet de Wit's lab also indicate effects of opioid drugs on social reward responses.)

d) The following claim is surprising given the abundance of behavioural economics and facial EMG/facial emotion recognition studies in the field: "human research has struggled to adopt an operationalization that resembles the one used in animal research, e.g. measuring behavior and facial reactions instead of relying on subjective verbal report". Could a better explanation be that researchers testing human participants have much less precise ways of altering and measuring activity of opioid, dopamine and other neurotransmitters?

e) fEMG results for positive stimuli often show *either* zygomaticus increases *or* corrugator decreases, in fact Ree et al., Mayo et al. and Bershad et al. all found only corrugator decreases during touch. A reanalysis of the data from Pawling et al. available at PloS ONE suggests that the zygomaticus effect no longer meets statistical significance when only the time receiving touch is considered, and the interval after touch is removed. Please rephrase the Introduction section that currently indicates that both muscle potentials are altered in such gentle touch paradigms.

28) The end paragraph of the Discussion provides clarity to some issues noted above. It would be helpful to provide this clarity early on as the comments in the early part of the Discussion section give an impression of overgeneralizing some of the findings.

29) In the interpretation of the statistical findings please provide careful descriptions of the data and not just the p values? This includes addressing the following:

a) Please go through the manuscript carefully to avoid terminology such as "did not differ", when what is meant is "did not significantly differ". It is only correct to say that two measures do not differ if they are identical.

b) It is recommended to moderate the statement that an influence of drug via mood can be excluded.

c) P values above the α level are not accurately described as "trends"; instead please describe the pattern of the data and state clearly that the required significance level was not reached.

d) “Importantly, this finding cannot be explained by mouth movements that might have occurred during food delivery (although instruction to swallow followed the Delivery window), as statistics did not reveal an interaction with the factor Condition.”: a null effect (interaction) is interpreted as evidence that a potential confound did not influence the results. This is inappropriate with frequentist statistics.

30) Please replace the CS and ZM abbreviations with corrugator and zygomaticus. CS is commonly used to denote conditioned stimulus thus causes some confusion, especially when used in the graphs. Further, RewardType gives the impression that it denotes drink versus touch, rather than reward level. In Discussion paragraph six, it is suggested to clarify the announcement of possible reward or similar, since participants only received the announced rewards they chose to work for. Further, please edit phrases including "the fact that" when describing the results.

31) Please provide acknowledgement of limitations in the study, which certainly exists e.g. cross-sectional design (rather than within-subjects). This can be noted when discussing the interpretation of the findings.

[Editors' note: further revisions were suggested prior to acceptance, as described below.]

Thank you for re-submitting your article "Dopaminergic and opioidergic regulation during anticipation and consumption of social and nonsocial rewards" for consideration by *eLife*. Your article has been reviewed by three peer reviewers, and the evaluation has been overseen by a Reviewing Editor and Christian Büchel as the Senior Editor. The following individuals involved in review of your submission have agreed to reveal their identity: Siri Leknes (Reviewer #1); Guillaume Sescousse (Reviewer #2); Diego Pizzagalli and Yuen-Siang Ang (Reviewer #3).

While the manuscript has been improved, there are a few outstanding issues remaining.

Revisions:

The authors were very responsive and provided a thorough revision of their manuscript, which is a very solid and valuable contribution to the literature. However, there are still some concerns that should be addressed.

1) The corrugator and zygomaticus data is a strength of the manuscript, but the current manuscript emphasizes facial responses more than is warranted by the study design and the hypotheses. Please address the following:

a) Please specify the lack of significant reductions in subjective ratings of chocolate milk and stroking touch in the Abstract.

b) The phrasing in the Introduction still appears to indicate that animal studies provide the best answers, whereas human subjective report is less valuable. However, the animal studies are severely hampered by their use of an indirect measure of liking, which has not been generalized beyond food reward but instead their strength in animal models is the ability to manipulate specific circuits/neurons. Please edit the text to reflect these limitations and strengths in animal models

c) The numerous figures of fEMG data patterns do not seem warranted, i.e. do they present the answers to the most important questions? It seems as though all effects below p=0.1 are illustrated.

d) The comparable subjective wanting and liking ratings in the placebo, naltrexone and amisulpride are a main result and should be listed as such in the Abstract, Results and Discussion. Further, please clarify that face muscle responses to high-reward stimuli were also comparable after drugs. Perhaps equivalence testing or Bayesian stats could be used to address the credibility of these interesting null results?

e) Referring to reduced activity of a frowning muscle as an indication of “hedonic anticipatory pleasure” is unwarranted and likely to confuse readers with respect to what the data shows. The corrugator is not a good measure of pleasure. Similarly, the discussion of drug effects on “hedonic responses” to touch gives the impression that the drugs altered touch pleasantness, instead of a face muscle response, which was not measured in most other drug studies.

2) The presentation of the behavioral wanting data to illustrate the two significant two-way interactions is confusing. It's better to depict non-pooled data for each reward type and level, and use lines to indicate the significant interactions. It is challenging to determine from the figure whether there was an effect of drug for wanting of gentle caresses, for instance. Please address the following:

a) The authors now provide this information in the supplement. However, showing one chart per drug instead of one per reward type/level makes it hard to discern any drug-related differences. The x axis labeling in the supplementary file was also confusing. It is recommended to remove Figure 3 and replace it with the reorganized supplementary to clarify any drug differences.

3) Please change the terminology to accurately depict the data and statistics

a) If the authors use 0.05 as their α level it is incorrect to describe results above 0.05 as trends or marginal effects. The higher force levels should be described as non-significantly higher. Similarly, the legend of Figure 5 makes too strong a case given the data. 1. This terminology is statistically inappropriate and maintains some unnecessary ambiguity in the literature. Many experts in statistics have pointed out this problem in recent years (see e.g. Gibbs and Gibbs, 2015, https://academic.oup.com/bja/article/115/3/337/312358). It is encouraged that the authors remove these terms.

b) It is also recommended to use more accurate descriptors of group differences, given the between-groups design, i.e. use “higher” instead of “increased”, “lower” instead of “reduced” etc.

4) The Results section is very hard to read especially since there are multiple periods of time analyzed. Please explain in the beginning of each section what information should be gained from the analyzed data rather than providing some of this information is in the summary of EMG data. This would help the readers to better understand the data.

5) 50 mg naltrexone is not a low drug dose. It is the standard clinical dose and blocks 90-100% of mu opioid receptors. It is recommended that the authors revisit the PET literature and take into account that 50 mg naltrexone is considered sufficient to induce so-called “full” blockade of mu receptors, when interpreting their results.

---

## [Author Response]

Revisions:The reviewers conclude that the study is of interest for eLife and that additional experiments are not needed. However, the reviewers request substantial revisions focused on the analysis and interpretations of the study.1) The authors modeled their sample size on a previous study that showed small drug effects (Weber et al., 2016). Please clearly acknowledge the lack of power to detect small effects and provide interpretations of the results relevant to these small effects.

We have added the following paragraph at the end of the Discussion: “… the study may have suffered from a lack of power to detect small effects. We had modelled the sample size on a previous study using the same drugs and doses (Weber et al., 2016). However, Weber et al., 2016, only found relatively small drug effects, and several other studies have failed to show effects of a pharmacological modulation of the opioid or oxytocin systems on liking of affective touch (Ellingsen et al., 2014; Løseth et al., 2019). This reveals the difficulty of uncovering the neurochemical basis of reward processing in humans, and suggests that larger sample sizes should be used in future pharmacological studies to investigate the neurochemical bases of touch and other rewards.”

2) Please clarify why the main tests for the relationship between implicit and explicit responses were conducted in the full sample, rather than first being demonstrated only in the placebo group.

Similar, albeit smaller, effects are also found when using placebo participants only. However, we have now opted for removing these analyses all together, as they were also contained in the later analyses that included the factor Drug, and thus they were redundant.

Additionally, please clarify why the results are presented as pooled means instead of showing the mean and variance data for each drug, reward type and reward level.

Assuming that the reviewer is referring to Figure 3 in the previous version of the manuscript: EMG by Drug was not shown because the factor Drug was not included in these initial analyses, which had the goal to investigate the relationship between each muscle and wanting, effort, and liking. As written above, we have now removed these analyses, as they were redundant with the ones also containing the factor Drug.

On the basis of the interaction effects with p's lower than 0.05, it is concluded that "low rewards" are affected by opioid and dopamine D2/D3 antagonists. Please demonstrate that this result is indeed comparable for touch and taste rewards.

The LMM on effort included the 3 predictors Drug, Reward Level, and Reward Type. It resulted in a Drug X Reward Level interaction (Figure 3B), and there was no statistically significant 3-way interaction with Reward Type. Based on this, we conclude that there is not sufficient statistical evidence to suggest that this interaction differs between Reward Types.

3) Please clarify if side effects of the drugs were measured? For instance, nausea could affect responses to food stimuli.

We appreciate the reviewer’s question, and are sorry for not having reported this in the previous version of the manuscript. A self-report questionnaire had been used to measure nausea and 50 other possible side effects on a 4-point Likert scale (with the anchors 1 = “not at all” and 4 = “very much”). The side-effects questionnaire was filled out by participants at time of pill intake (T1) and three hours later (T2). This side-effects data was missing for one participant at T1 and for four participants at T2.

The side-effects did not differ between groups, neither at T1, nor at T2. We have added this information for nausea to Table 1 in the manuscript, and in Author response image 1 we show all the side effects across the 3 groups at T2.

**Author response image 1. respfig1:** Nausea and other possible side effects were measured through self-report (4 options: “not at all”, “slightly”, “moderately”, and “very much“) three hours after pill intake. All reported side effects were generally low, and did not differ between groups. The list of side effects (from top left to bottom right in the figure; original in German) was: headache; unrest; nervousness; belly ache; abdominal cramps; nausea; vomiting; joint/muscle pain; weakness; anxiety; thirst; swindle; dizziness; chills; sweating; tiredness; tears; tachycardia; chest pain; shortness of breath; diarrhoea; rush; itch; loss of appetite; increased energy; depressed; irritability; hallucinations; confusion; persecution mania; disorientation; tremor; eye irritation/swelling; light sensitivity; eye pain/fatigue; color vision deficiency; ear problems; ear pain; blushing; nasal problems; sneezing; coughing; increased yawning; flatulence; dry mouth; cold arms/legs; hot flashes; increased appetite; visual disturbance; speech disturbance; tinnitus.

4) Outlier removal is commonly applied with psychophysiological data, but not for VAS ratings. Please clarify why the outlier values were removed for these measures in the present study. Additionally, do the results differ with and without the excluded datapoints?

The goal of artefact removal was to reduce variability. We have followed the same procedure for outlier removal as in the previous publication with the same task (Korb et al., 2020). Differences in results with and without outliers are small at best, as can be appreciated by the forest plots comparing the models in Author response images 2, 3 and 4.

**Author response image 2. respfig2:** 

**Author response image 3. respfig3:** 

**Author response image 4. respfig4:** 

5) Please provide clarity into the details of the task and the outcomes:a) It appears that the 3 reward levels for food reward were milk, chocolate milk, and a 4:1 mix of milk and chocolate milk? Was the ranking of these 3 reward levels consistent across participants? It is plausible that some participants would prefer chocolate milk while others would prefer milk, and that the strength of this preference (i.e. how far apart liking ratings are) would widely vary between participants. It would be informative to report (in supplementary material) the pre-task liking ratings for the 3 reward levels, both for the food and touch rewards. In particular, it would be nice to show that these pre-task ratings did not differ between groups, as this would strengthen the interpretation of the Drug x Reward type interaction reported in Figure 2C as a genuine effect of amisulpride, rather than a mere effect of pre-existing differences between groups.

The 3 levels of food rewards were exactly as described by the reviewer. Participants indicated their preferences at the beginning of the experiment (see Procedure section for a description of the short task during which participants experienced and ranked all stimuli).

The variability in stimulus preferences was small and did not differ across Drug groups. We have conducted additional analyses, the results of which are described in the manuscript:

“In order to rule out eventual group differences that were not of interest, we conducted a series of statistical tests to verify the matching of the three groups.

The three drug groups did not differ in rankings of rewards before the main task, as shown by the absence of a significant Drug X Reward Level interaction for both food and touch rewards (Figure 2). Only a significant main effect of Reward Level was found for food (Χ2 (2) = 78.1, *p* <.001) and for touch rewards (Χ2 (2) = 115.71, *p* <.001), confirming the expected pattern of preferred food rewards (milk with greater chocolate content being preferred to milk with lower chocolate content), and of touch rewards (slower caresses being preferred to faster caresses).”

Please see Figure 2 for participants’ initial reward preferences by Reward Type and Reward level.

Moreover, we dealt with eventual differences in participants’ preferences by (1) adapting the task to participants’ initial choices, and (2) using for the analyses of the EMG ratings of wanting and liking on a trial-by-trial basis, which allowed us to even track changes from initial preferences.

b) Expression "if squeezing was not sufficient" and "insufficient effort" might be misleading as it implies that a minimum amount of squeezing (i.e. a threshold) was required to obtain a reward. Please clarify that there is a linear (i.e. continuous) relationship between squeezing and the probability of gaining a reward.

We have modified the manuscript to clearly indicate that there was a linear relationship between effort and probability of obtaining the announced reward.

Legend of Figure 1: “The probability of obtaining the announced reward was determined linearly by participants’ hand-squeezing effort, which was indicated in real-time. Participants knew that they would obtain the announced reward if they reached the top of the displayed vertical bar, which corresponded to their previously measured maximum voluntary contraction (MVC). The gained reward (which was either the one announced at the beginning of the trial, or – in the case of lower probability due to less squeezing – the leastliked “verylow” reward) was then announced and delivered.”

“In the main task, the level of reward (high, low, verylow) received in each trial depended on both the announcement cue at the beginning (high or low) and the force exerted to obtain it (verylow rewards were only obtained when participants’ exerted low effort, which linearly converted into low probability to obtain the announced reward).”

“which could be high, low, or – if insufficient effort had been exerted – verylow (the greater participants’ effort, the higher the probability of obtaining the announced reward)”.

c) Please clarify it there were as many high reward as low reward cues.

Half of the trials contained cues for high rewards, and the other half for low rewards. We have modified parts of the manuscript, including legend of Figure 1, to make this clearer: “In the main task (here depicted), the highest-ranked (“high”) reward was announced in half of the trials, and the second-highest ranked (“low”) reward was announced in the other half of trials”

The number of high and low (or verylow) rewards received could vary between participants. As indicated in the text: “In the main task, the level of reward (high, low, verylow) received in each trial depended on both the announcement cue at the beginning (high or low) and the force exerted to obtain it (verylow rewards were only obtained when participants’ exerted low effort, which linearly converted into low probability to obtain the announced reward). The number of trials in which high, low, and verylow rewards were obtained did not differ across groups. Only a significant main effect of Reward Level was found (F(2, 763) = 27.84, p <.001), due to a greater number of high (M = 33.07, SD = 4.97) than low (M = 29.85, SD = 6.03) and verylow (M = 17.14, SD = 8.99) trials received, across all three drug groups and both reward types.”

6) Based on previous literature, when is the typical peak of maximal efficiency for amisulpride and naltrexone, and does this peak coincide with the 3h delay used in the current study?

We have added the following to the Procedure:

“Amisulpride reaches a first peak in serum after one hour, and a second (higher) peak after approximately four hours. The elimination half-life is of 12 hours (Rosenzweig et al., 2002). At doses of 400 mg or higher, amisulpride acts as a postsynaptic D2/D3 receptor antagonist, and thus results in reduced dopaminergic action (Racagni et al., 2004; Schoemaker et al., 1997). Naltrexone reaches maximal concentration in plasma after one hour, has an elimination half-life in plasma of approximately four hours, and is completely cleared from plasma after 96 hours (Meyer et al., 1984). Importantly, up to 90% of mu-opioid receptors in the brain remained blocked by naltrexone after 48 hours, and partial receptor blockade could be shown up to 168 hours (Lee et al., 1988).”

The drug doses and the time delay between drug ingestion and starting of the main task were chosen based on a previous study that combined 50 mg of naltrexone and 400 mg of amisulpride (Weber et al., 2016). They result in comparable neurochemical responses, that were well tolerated in healthy participants.

7) There are 5 dependent variables (effort, wanting and liking ratings, EMG of CS and ZM muscles) tested at 4 different times (Pre-effort anticipation, Post-effort anticipation, Delivery and Relax), as a function of 3 main conditions (Drug, Reward type, and Reward level). As a result, there is a large number of statistical tests being performed, thus a correction for multiple comparisons would be warranted (see e.g. Cramer et al., Psychon Bull Rev 2016).

We have taken the following steps to limit the risk of committing Type I errors, while retaining sufficient power and mitigating the risk of Type II errors.

First, we used linear mixed models (LMMs) including random slopes. This type of analyses is more conservative (and thus reduces the risk of Type I errors) than ANOVA.

Second, we now control for the false discovery rate (FDR) associated with multiple testing of the EMG data using the Benjamini-Hochberg method (Benjamini and Hochberg, 1995). This correction is also recommended by Cramer et al., 2016. We have indicated so in the Analyses section of the Materials and methods. To be noted that, as a result of the correction (and of the inclusion of the covariate Block to control for changes over time), the Liking X Drug interaction for the zygomaticus muscle changed from significant (p = .02) to now marginally significant (p = .09).

8) Due to the repetitive nature of the ratings on every trial, while there's no incentive for the participants to provide faithful ratings, the current procedure may run the risk of producing stereotyped ratings with little variability. Could the authors demonstrate that this is not the case? Further, there are possible satiety effects (i.e. ratings going down gradually, especially for the food reward). Please clarify if this occurred.

Figure 2—figure supplement 1 shows wanting, force, and liking of the 3 Reward Levels (high, low, and for liking also verylow) across the 2 Reward Types (Food, Touch) and the 3 Drug groups. Wanting, force, and liking decreased over the course of the experiment, in agreement with the literature. The fact that ratings and force were not constant throughout, seems to contradict the reviewer’s suggestion that ratings were stereotypical and non-faithful.

Importantly, we have controlled for changes over time by including the factor Block (recoded by order and reward type) as a covariate to all analyses.

9) The authors have demonstrated a dissociation between drug effects on wanting and liking measures. However, Point 2 (“does the neurochemical basis of human reward processing differ for social and nonsocial rewards”) is overstated. The results from one social and one nonsocial stimulus type is insufficient to allow generalization to a class of rewards. Moreover, several drug studies have reported being unable to alter liking and /or wanting measures of stroking touch. This includes morphine, naltrexone, tryptophan and oxytocin. To the reviewers' knowledge, the only drug that alters perception of stroking touch is MDMA. Please provide discussion of the current understanding of the difference between drug effects on food and touch reward.

We agree that our conclusions are limited to the here-used types of rewards, and that more research is required to verify them in different social and nonsocial rewards. We have added this limitation to the Discussion: “The current study is characterised by a number of limitations. First, only two types of stimuli (food and touch) were used to define social and non-social rewards, and only two neurochemical systems (dopaminergic and opioidergic) were challenged. […] Future studies should therefore broaden the neuropharmacological investigation of social vs. nonsocial reward processing, by using other compounds and different rewarding stimuli, to allow the generalizability of the findings to other social and nonsocial rewards, and to better understand their neurochemical basis.”

We have referred to the finding that MDMA administration increases ratings of pleasantness and zygomaticus activation to touch. This work is clearly relevant, as MDMA alters the levels of several neurotransmitters, such as serotonin, dopamine, noradrenaline, and possibly oxytocin: “This is also suggested by the finding of higher pleasantness ratings and greater zygomaticus activation to touch after administration of 3,4-methylenedioxymethamphetamine (MDMA), a drug that modulates serotonin, dopamine, and possibly oxytocin levels (Bershad et al., 2019; de Wit and Bershad, 2020).”

We have discussed our findings in relationship to existing drug challenges studies on stroking touch: “To the best of our knowledge, this is the second study (after Bershad et al., 2019) to report a pharmacological modulation of hedonic responses to experienced touch (for a trend like effect see also Case et al., 2016). Major differences in our study compared to previous work (Ellingsen et al., 2014; Løseth et al., 2019) are the delivery of touch with the hand instead of a brush (Ellingsen et al., 2014 included touch by hand but wearing a glove), and the initial selection of the preferred touch speed by the participants. Regarding the delivery human touch vs. brushing, it is possible that the social saliency of the touch stimuli delivered through direct skin contact was enhanced, allowing us to detect subtle effects of the drug. Regarding the selection of the preferred touch speed, even if the majority of our subjects selected the slower speed as the preferred one, inter-individual differences were observed, and the implementation of a task that could account for those, may have helped to detect the effect of the drug. Further studies should investigate how such factors modulate drug responses to perceived affective touch.”

10) The decreased effort levels observed for food rewards in the amisulpride and naltrexone groups compared to the placebo group convincingly argue in favor of a role of the dopamine and opioid systems in motivation. However, and in contrast to the hypothesis stated in the Introduction, the same pattern wasn't observed for touch rewards. This discrepancy limits the generalizability of the conclusion, and warrants discussion. This also applies to the effect of amisulpride and naltrexone on CS muscle contraction (Figure 4): the fact that these effects are restricted to food reward limits the generalizability of the conclusion that the dopamine and opioid systems are involved in anticipated pleasure.

We fully agree, that the results on effort and partly also on the EMG were either stronger for food, or only present for food trials. We have indicated this on several occasions in the Discussion, including the following section:

“Although the magnitude of the two types of rewards in terms of subjective ratings and effort was carefully matched (Korb et al., 2020), most drug effects were either stronger or restricted to food trials, as indicated by significant Drug x Reward Type interactions for measures of effort to obtain the announced reward, and for corrugator activation during PreEffort anticipation. […] This point however requires further investigation through brain imaging, or through more direct measures of brain activity, in addition to pharmacological challenges tailored to investigate the role of different neurochemical systems in the processing of social vs. nonsocial rewards.”

11) Additionally, for Figure 4, please clarify why reward magnitude was not included as a fixed factor similar to other analyses? Might dopamine and opioid antagonism be dissociable effects in terms of their effect on reward magnitude? Similarly, for effort analysis.

Reward magnitude was not entered in the statistical models for EMG analyses. Instead, the EMG data was analysed using the continuous predictors consisting in trial-by-trial ratings of wanting and liking, as well as trial-by-trial exerted effort. We consider this approach as superior to using the categorical predictor Reward Level (with 2 or 3 levels depending on the time window). Former Figure 4, now Figure 4D, only shows the reward type (food, touch) X drug interaction, because the wanting X drug and the 3-way reward type X wanting X drug interactions were not significant. Thus, to answer the reviewer’s question, we do not have any statistical grounds to argue that dopamine or opioid antagonists have dissociable effects on corrugator activation in the pre-effort anticipation period, depending on reward magnitude (i.e. reward level).

Behavioural data, on the other hand, was analysed with the categorical predictor RewardType, because it is not possible to do otherwise (i.e. you cannot use continuous ratings as both dependent variable and predictor).

12) In the Discussion the statement "the results speak for a genuine alteration of the incentive salience of the low reward, whose rewarding value approached that of the verylow reward, due to the pharmacological manipulation" might be overstated and not supported by the findings. Subjects were never asked to exert effort to pursue verylow reward, hence, the rewarding/motivational value of verylow reward is unknown and it is inaccurate to conclude that the low reward's "rewarding value approached that for verylow reward". Moreover, Figure 2C showed that subjects liked the low rewards much more than verylow rewards; although this is not motivational salience, it might be reasonable to expect them to be associated to some degree. An elegant way to extract these motivational values would be to apply computational modelling, which has been conducted in other similar effort-based decision-making tasks.

We agree that we cannot, based on effort alone, claim that the reward value of low rewards approached that of verylow rewards in the 2 drug groups. We have modified the corresponding paragraph and now discuss the finding in the following terms: “Notably, and differently from our original hypothesis, the effect was most pronounced for the second-preferred (low) rewards, as indicated by a significant Drug x Reward Level interaction. A possible explanation for this finding is that our food stimuli did not vary on caloric content (i.e. the three reward levels were carefully matched for fat and sugar). Therefore, individual preferences were derived from other mechanisms than energy value, possibly leading to a different effect of the drug on food reward processing (Barbano et al., 2009; Salamone et al., 2007). Another possible explanation is that high rewards are less susceptible to changes in their incentive salience, when more options are available.”

We also agree that it would be worthwhile to apply computational modelling to this data, which however is not part of our current skill set, and therefore cannot be included into the paper at this time. A sentence highlighting the added value of using computational models in pharmacological challenge studies on reward processing has been added to the Discussion: “Furthermore, computational approaches will most likely be useful to reveal hidden psychological states subtending motivation and experienced pleasure, allowing to refine how drug administration acts on these two components (Meyer et al., 2019).”

13) In the Discussion the findings do not appear to "corroborate the hypothesis that the dopaminergic system underlies the motivational but not the hedonic component of rewards, while the opioidergic system underlies both." Figure 2C shows that the liking result for placebo is similar to naltrexone and not amisulpride. Using placebo as the reference for comparison, the findings appear to show instead that dopamine antagonism has an influence on increasing hedonism, but opioid antagonism has no effect, which is surprising in the context of existing literature.

The new analyses (with block as covariate) did not confirm the initial findings of Druginduced differences in liking (Drug X Reward Type interaction not significant, *p* = .4). At the behavioural level, amisulpride and naltrexone only affected effort levels. We therefore feel more confident in claiming such differential effects of opioidergic and dopaminergic drugs on wanting vs. liking.

14) In the Discussion please clearly state how the results align or do not align with the hypotheses. For instance, the discussion of the naltrexone group's zygomaticus “liking” responses gives the appearance as if the result was expected. However, it is surprising that opioid blockade (1) did not reduce subjective liking for any stimuli, not even high-calorie foods, and (2) appears to have reduced not zygomaticus activity in general, but more specifically its association with ratings of liking.

We have added the following section to the Discussion: “Interestingly, drug effects were found on effort levels and fEMG, but not on subjective ratings of wanting and liking. While this may come as a surprise, it is in line with several previous studies, which have reported either null or weak effects of pharmacological interventions on pleasantness likings of affective touch (Case et al., 2016; Ellingsen et al., 2014; Løseth et al., 2019; Trotter et al., 2016). It thus appears that more implicit measures are required, to succesfully observe drug-induced changes in wanting and liking of at least these two types of rewards.”

The reviewer is right in pointing out that we had specified a reduction of hedonic facial reactions (i.e. less zygomaticus, more corrugator). Specifically, in the Introduction, we say: “subjective ratings of liking, and hedonic facial reactions during reward consumption, were expected to be reduced after administration of the opioid antagonist naltrexone, compared to placebo”. However, we realised that we had not well formulated this hypothesis, even though we had written about it elsewhere in the text. Indeed, the literature frequently suggests the effect of opioidergic modulation is observable on the most palatable food or most attractive faces (Eikemo et al., 2016; Smith and Berridge, 2007). We have therefore specified better the initial hypotheses, and we refer now to effects in “the most preferred rewards”.

The analyses of fEMG data performed with subjective ratings as continuous predictors allowed us to assess two things: (1) overall changes in the activity of the muscle (main effect) and (2) changes in the association between subjective ratings and facial activity (which is a proxy for assessing the effects of a drug on different levels of the reward). We observed that naltrexone indeed altered this association, with zygomaticus activating less to preferred than non-preferred rewards. We now discuss these findings based on the adjusted hypotheses: “Notably, and differently from our original hypothesis, the effect was most pronounced for the second-preferred (low) rewards, as indicated by a significant Drug x Reward Level interaction.”

15) Please provide in the Discussion an interpretation of the effects of naltrexone and amisulpride at the doses chosen.

We have added to the Discussion a section on limitations of this study. There, we also discuss the findings in relation to the drug doses used: “Second, we used low drug doses (50 mg of naltrexone, and 400mg of amisulpride). […] Upon availability of drug compounds that more strongly modulate the opioidergic and dopaminergic systems with minimal side effects, future studies should, however, explore dose-dependent changes in human reward processing.”

16) The authors suggest that the CS and ZG measures are complementary, but perhaps they are actually more different than similar? CS assesses frowning, which indicates negativity, whereas ZG measures smiling, which indicates positivity. CS is a proxy for negative valence axis, while ZG measures only positive coordinates on the valence axis; neither is a continuous assessment across the whole spectrum from the negative to positive end. This has implications for the interpretations in the study. For example, the authors state that "greater frowning during reward anticipation might be interpreted as a reflection of less anticipated pleasure", but this could be more appropriately interpreted as greater frustration, which is different from positive anticipation (see e.g. Bremhorst et al., 2019, PMID: 31848389). To further this point, if CS and ZG are indeed complementary measures, one would also expect ZG effect during the Pre-Effort anticipation of food – but that is not seen. Similarly, for reward delivery, while there is a CS effect, there is no ZG effect. What is the association between the two measures? A more in-depth discussion would help to improve the interpretations of the study.

The reviewer seems to raise two points: first, whether each muscle is valence specific (i.e. zygomaticus = positive, CS = negative) and second, whether the two muscles are complementary.

Regarding the first point, we disagree with the view that the zygomaticus and corrugator only indicate, respectively, positivity and negativity. Instead, pleasure and positive affect typically activate the zygomaticus and relax the corrugator, while negative affect activates the corrugator and relaxes the zygomaticus. This has been shown in response to pictures, sounds, and words covering a large valence spectrum from negative to positive (Lang et al., 1993; Larsen et al., 2003). There is no reason to expect different muscle responses to rewards than other affective stimuli. Importantly, the inverse linear relationship between corrugator activation and valence is not specific to negative affect, such as frustration. On the contrary, Larsen et al., 2003, found that, “positive and negative affect have reciprocal effects on activity over corrugator supercilii” (Abstract), and “even when negative affect is held constant at minimal levels, positive affect inhibits activity over corrugator supercilii”. In other words, corrugator activation does not only increase in the range of neutral-tonegative affect, but relaxes even more in the neutral-to-positive affect range. We have added the references of Lang et al., 1993 and Larsen et al., 2003 to the manuscript.

Certainly, it remains unclear why the drug effects on corrugator and zygomaticus were so different. We have added the following to the Discussion: “Interestingly, while the corrugator showed a general increase due to drug administration, zygomaticus activity was only affected in its relationship to subjective pleasure, and not in terms of overall activation. This differential pattern suggests that the two muscles, despite both tracking changes in hedonic value (Lang et al., 1993; Larsen et al., 2003), do not necessarily behave in a complementary, but rather in an independent way. This might also explain the heterogeneity of findings in previous studies, which often report an effect of reward valence on only one of the two muscles.”

Although it is true that our task could have induced some frustration when the cued reward was not obtained, this frustration must have been mild at best, as even in those trials a good-tasting reward (and not a punishment) was delivered. Moreover, the earliest time point at which frustration could have appeared in each trial is after (and not before) the physical effort, i.e. from the Post-Effort anticipation phase onwards. Therefore, the inverse linear relation between wanting/effort and corrugator activation in Pre-Effort anticipation period cannot be explained by frustration due to not obtaining the cued reward (since this was decided later in the trial). Alternatively, frustration may have increased over the course of the experiment, due to the repeated administration of a limited number of rewards, and could therefore have affected all time windows of analyses, including the Pre-Effort anticipation and Relax phases, for which we found changes in the corrugator. This is unlikely to have been the case, however, since in the now-reported analyses we controlled for the factor Block (and thus eventual time- and repetition-induced effects of habituation and fatigue).

As to the zygomaticus muscle, it is true that previous research has reported not only a linear, but also a quadratic effect of valence, reflecting greater zygomaticus contraction for both extremely positive and extremely negative affect (Lang et al., 1993; Larsen et al., 2003). Since we only administered positive, rewarding stimuli, we believe that it is unlikely that our effects on the zygomaticus reflected increased activation to very negative valence. However, it is true that we cannot fully exclude the possibility that the inverse relationship of valence and zygomaticus activation, which was found for the naltrexone group during reward delivery, is due to smirking or disgust reactions to the less-liked rewards. This remains unlikely, however, because the number of trials with high, low, and verylow rewards did not differ between drug groups, and because the ratings of wanting and liking of these rewards did not differ across drug groups, and finally because the change of ratings of liking over time was comparable across drug groups (as shown by results remaining when controlling for the factor block, see also Figure 2—figure supplement 1).

To explain why we interpret the reversed liking-to-zygomaticus pattern in the naltrexone group as a reduction of hedonic facial responses to the most liked rewards (and not as a reflection of negative facial responses to the least liked rewards), we have added the following paragraph to the Discussion:

“Does zygomaticus activation for less liked rewards in the naltrexone group reflect negative facial reactions? […] Nevertheless, it is important to keep in mind that the drug effects on zygomaticus activation during reward delivery were only marginally significant, and should therefore be interpreted with caution.”

17) How do the authors interpret the finding that during reward delivery, the ZM-liking slope for amisulpride did not differ from placebo nor from naltrexone? Comparisons between naloxone and placebo were interpreted to mean opioid antagonism reduced smiling, but no discussion was made for dopamine antagonism.

In the previous version of the manuscript, we had not discussed the effects of the dopamine-receptor antagonist amisulpride on ZM activation slope, because precisely the slope was not significantly different from the slope for the other groups. However, reanalysis (with FDR-corrections and the covariate Block) now result in a slightly different picture. As written in the text: “the interaction was explained by greater liking resulting, as expected, in greater zygomaticus activation in the placebo group (b = 5.74), and to a lesser extent also in the amisulpride group (b = 0.56). In contrast, the opposite pattern was found in the naltrexone group, with a negative slope (b = -7.86) that was significantly different from the placebo group (p = .006) and marginally significantly different from the amisulpride group (p = .09). The amisulpride and placebo group did not differ between each other (p = .3).”

Thus, as in our previous discussion, naltrexone seems to alter and reverse the typical link between liking and smiling, but amisulpride does not. In fact, the slopes for amisulpride and placebo are not significantly different from each other. But the slope of amisulpride, which is positive like the one for placebo, is marginally different from the naltrexone group.

18) It is unclear from the main text or Materials and methods what feature of the force-time curve is used as the proxy of effort exerted – later in the legend of Figure 3—figure supplement 1, it is stated that "exerted force is the maximum value reached in the 4-sec period". If that is correct, please state this more clearly within the manuscript and also explain, what is the rationale for adopting the peak in a single infinitestimal timepoint within the 4-sec window as the measure of exerted force? Based on experience with hand-dynamometers in physical effort paradigms, it is not easy to control the force one intends to exert. Hence, there is typically an initial sharp spike in the force-time curve where the peak occurs (but not necessarily reflecting what the subject intends to exert), before stabilizing to a plateau that more accurately reflects the subjects intended force exerted. Thus, it is more typical in other studies to require subjects to sustain their force for say, 2s out of a 4-s window, and then take that sustained force level as a proxy of effort exerted.

As written in subsection “Procedure” the force was displayed as percentage of the MVC, averaged with a sliding average of one second. The maximum force was also taken based on this sliding average. Therefore, we did not take an infinitesimal timepoint of exerted force, but the average of 1 second. This value was chosen based on pilot studies, and was also used for our previous publication (Korb et al., 2020).

19) Further, how is force exerted converted to probability? What constitutes as insufficient effort? The probability component seem an unnecessary manipulation thus what was the authors' rationale for rewarding probabilistically, rather than delivering the reward as long as sufficient effort is exerted?

Force (i.e. maximum in a 4-sec period, as percentage of MVC, and after passing a 1second sliding average) was converted linearly into probability to obtain the announced reward. An insufficient effort thus constitutes either no effort at all (resulting in zero probability of receiving the announced reward, and conversely in 100% probability of obtaining the verylow reward), or a relatively low effort (e.g. 10% of the MVC only results in a 10% probability of receiving the announced reward).

We chose to reward probabilistically based on effort, as we wanted a continuous measure of wanting. Had we instead delivered a reward based on a specific effort threshold, we probably would have obtained less variability in the amount of exerted effort between trials. Furthermore, the conversion to probability implicated a certain amount of uncertainty, which allowed us to induce an anticipated response in the post-effort anticipation (otherwise completely explained by the pre-effort anticipation).

A linear conversion of effort into the probability of obtaining a reward might not be the most frequently used paradigm, but it has been used before in the relevant literature (Lopez-Persem et al., 2017).

20) A substantial number of subjects have a mean force exertion much higher than 100% of their MVC, especially in the amisulpride group (Figure 3—figure supplement 1) – does that mean the baseline measurement of MVC is not accurate since it is not actually maximum? Alternatively, it might also reflect issues associated with taking peak at the spike of the force-time curve as the measure of force exerted.

There is unfortunately no perfect way of calibrating the dynamometer, explaining why forces > the MVC can sometimes be obtained. We have explicitly measured the MVC before explaining the task to participants, in order to keep them as naïve as possible about the role that their effort would later play in the task. Nevertheless, participants may sometimes press less than their maximum force during MVC measurement. For the experimenter, it is very hard to detect and correct this. This is a recurrent problem in this type of research.

Importantly, trials with effort equal to or greater than 100% of MVC had the same probability (i.e. 100%) of obtaining the announced reward, and this was explained to participants.

Again, we took the peak of effort in each trial but only after averaging with a 1-second sliding average.

21) The authors do not account for the effect of fatigue in their analyses, which is presumably substantial given the large number of trials they have to squeeze the hand-grip dynamometer. The authors might include a factor of block?

We have included the factor block (recoded into first and second block for each reward type) as a covariate to all analyses. Doing so, we made sure that the results remained significant while controlling for the effects of habituation and fatigue. We have also added Figure 2—figure supplement 1 to show wanting, liking, and effort across trials, Reward Types, and Reward Levels.

22) Please provide rationale behind the choice of variables in the LMM, which includes all fixed factors and their interactions, as well as subject random intercepts and random slopes for all within-subject factors and their interactions. This appears as overparameterization. It is not easy to detect overparameterization since each random effect uses only one df, but the estimation of variance among small numbers of groups can be numerically unstable. The approach of fitting a model as complex as possible is problematic as it makes it very easy to fit a model too complex for the data. If the model fails to converge, that is fine since inference is not possible. But if the model "works", then that risks false inference (see e.g. Zurr et al., 2010).

We have chosen the fixed effects based on the experimental design and hypotheses (for example, we explicitly wanted to investigate if EMG varies by Reward Type, Drug, and ratings of wanting/liking and effort). We have put all fixed effects as random slopes, as this is often advised (Barr et al., 2013). According to simulations, fitting a maximal model does indeed have a cost, which however is not the increase of Type-I errors, but rather the loss of power (Matuschek et al., 2017).

23) The authors considered only one possible model. From a statistical point of view, however, the full range of possible interactions, from the full model (including all possible interactions between factors) to just the main effects, should be considered and the best model analyzed according to e.g. AIC/BIC.

We have designed the experiment to test the effects of several predictors in interaction, and have therefore fitted models that included them all – up to a limit, because otherwise the models do not converge. We favour a theory/hypothesis-driven approach, while the reviewer seems to suggest a more data-driven approach. As far as we are aware, both are legitimate procedures.

24) Please provide more details on how the LMM analyses were conducted. For example, did they use ML or REML, and why?

As we have now specified in the text (under Materials and methods, Analyses), LMMs were fitted with REML, as it provides unbiased estimates of the random effect variance-covariance matrix (https://en.wikipedia.org/wiki/Restricted_maximum_likelihood), and it is the default used by both lmer() and lmerTest(). Using the anova() function to compare models would automatically refit them with ML.

25) Please provide tables denoting all the parameter estimates, SE, F-statistic, p-value etc. for each LMM.

We have added the tables for all the statistical models to the supplementary material (using the function tab_model from the package sjPlot). Please be aware that these do not include F values, dfs, and p values as described in the manuscript – those were obtained with the function anova(), as stated in the manuscript. Please be also aware, that the pvalues for fEMG provided in the manuscript were corrected for multiple comparisons using Benjamini-Hochberg FDR, while tables include uncorrected p values.

26) In Results paragraph one: There are 80 trials in total across four blocks, but the total number of rewards obtained do not seem to add up to close to 80? Please clarify that the mean rewards reported are collapsed across condition.

We thank the reviewer for pointing out that we had erroneously provided the number of trials by Reward Type and Reward Level (because the statistical model also included the predictor Reward Type). We have now modified the corresponding text passage accordingly. To clarify: there were 80 trials per subject, separated in 4 blocks of 20 trials each. Of these half (40) were social rewards, and the other half (40) included food rewards. Each reward condition included half (20) of the trials announcing a high reward, and the other half (20) announcing a low reward.

Across both Reward Types, the average numbers of high, low, and verylow rewards were, respectively, 33.07, 29.85, and 17.14.

Additional major comments to improve the clarity of the study:27) The Introduction is very long. A more concise Introduction would improve the overall clarity.

The Introduction has been considerably shortened.

Further, please address the following in the Introduction:a) Wanting and liking have been linked primarily to dopamine and opioids, but it's useful to extend this rule of thumb to encompass evidence on endocannabinoids, orexin and even benzodiazepines in enhancing liking responses to a comparable degree as mu, kappa and delta opioid microstimulation. This perspective should also be considered in the conclusion section which should be rephrased to avoid the impression that you have tested every relevant neurotransmitter.

We have extended the literature evidence on other neurochemicals’ effect on liking and wanting both in the Introduction:

“On the other hand, (facial) hedonic reactions to sensory pleasure are amplified by opioid, orexin and endocannabinoid stimulation of various “hedonic hotspots” of the brain, including the nucleus accumbens (NAc) shell and limbic areas such as insula and orbitofrontal cortex (Berridge and Kringelbach, 2015).”

and in the Discussion:

“From the animal and human literature, we however know that other systems (e.g. endocannabinoids, orexin, benzodiazepine, etc.) also contribute to the motivational and hedonic components of reward processing (Berridge and Kringelbach, 2015). Future studies should therefore broaden the neuropharmacological investigation of social vs. nonsocial reward processing, by using other compounds and different rewarding stimuli, to allow the generalizability of the findings to other social and nonsocial rewards, and to better understand their neurochemical basis.”

b) Please provide an explanation for the interpretation that opioid-induced increases in motivation for high-calorie foods as evidence that opioids primarily affect wanting through liking.

After rewriting the Introduction, the sentence has been removed.

c) Please consider changing to past tense when reviewing the evidence from single/small numbers of studies. Also, the effects reported in Chelnokova et al., 2014 on wanting were not explicitly subjective, rather a count of button presses which prolonged or shortened the duration an image was shown on the screen. Other relevant references on opioids and liking/wanting in humans include Gospic et al., 2008, Psychopharmacol; Atlas et al., 2013; Eikemo et al., 2017; Chelnokova et al., 2016, SCAN. Importantly, Gospic, Atlas and colleagues tested both men and women, whereas the cited evidence from Leknes' lab and Büchel's erotic image study were conducted exclusively in men. (Note that the affective image stimuli used by Gospic, Atlas and colleagues could also be described as social, and studies from Harriet de Wit's lab also indicate effects of opioid drugs on social reward responses.)

We thank the reviewer for the suggested references. We have added them in the Introduction with the exception of Gospic which is only used neutral and negative pictures:

“Administration of µ-opioid receptor agonists in heathy individuals has been associated with changes of subjective feelings and motivational responses to different type of rewards, as indicated for example by increased pleasantness of the most palatable food option available (Eikemo et al., 2016), increased effort to view, and liking of the most attractive opposite sex faces (Chelnokova et al., 2014), increased preference for stimuli with high-reward probability (Eikemo et al., 2017), and enhanced emotional ratings to positive and negative images (Atlas et al., 2014). Furthermore, administration of the non-selective opioid receptor antagonist naloxone to healthy men decreased subjective pleasure associated with viewing erotic pictures and reduced the activation of reward related brain regions such as the ventral striatum (Büchel et al., 2018).”

d) The following claim is surprising given the abundance of behavioural economics and facial EMG/facial emotion recognition studies in the field: "human research has struggled to adopt an operationalization that resembles the one used in animal research, e.g. measuring behavior and facial reactions instead of relying on subjective verbal report". Could a better explanation be that researchers testing human participants have much less precise ways of altering and measuring activity of opioid, dopamine and other neurotransmitters?

We agree with the reviewer that the use of facial EMG is abundant in research addressing emotional reactions to different types of stimuli, and in the context of reward processing. We have reformulated the paragraph and hopefully made clear that the above-mentioned claim refers to pharmacological studies on human reward. The text in the Introduction reads as follow:

“In spite of the progress made, human and animal research about the neurochemical regulation of reward processing remain difficult to compare, as human pharmacological studies have struggled to adopt translational paradigms and an operationalization of reward that resembles the one used in animal research, i.e. measuring decision utility and experienced utility in the same task, providing primary rewards on a trial-by-trial basis, and/or using objective hedonic reactions to consumed rewards, instead of relying on subjective verbal report (Der-Avakian et al., 2016; Pool et al., 2016). Hedonic facial reactions, the “gold standard” in animal research on the neurochemical regulation of reward processing, seems a promising research tool in this regard. Recently, the use of facial electromyography (fEMG) has gained increased attention in the context of human reward processing. Results suggested that human adults relax the corrugator muscle (involved in frowning), and to a lesser extent activate the zygomaticus muscle (involved in smiling), during both the anticipation and the consumption of different types of pleasurable stimuli, although differences between types of rewards exist (Bershad et al., 2019; Franzen and Brinkmann, 2016; Korb et al., 2020; Mayo et al., 2018; Pawling et al., 2017; Rasch et al., 2015; Ree et al., 2019; Sato et al., 2020; Wu et al., 2015).”

e) fEMG results for positive stimuli often show either zygomaticus increases or corrugator decreases, in fact Ree et al., Mayo et al. and Bershad et al. all found only corrugator decreases during touch. A reanalysis of the data from Pawling et al. available at PloS ONE suggests that the zygomaticus effect no longer meets statistical significance when only the time receiving touch is considered, and the interval after touch is removed. Please rephrase the Introduction section that currently indicates that both muscle potentials are altered in such gentle touch paradigms.

We thank the reviewer for pointing out that most studies have found corrugator relaxation for pleasant touch, while zygomaticus increase for touch appears more rarely. We have modified the Introduction accordingly:

“Results suggested that human adults relax the corrugator muscle (involved in frowning), and to a lesser extent activate the zygomaticus muscle (involved in smiling), during both the anticipation and the consumption of different types of pleasurable stimuli, although differences between types of rewards exist (Bershad et al., 2019; Franzen and Brinkmann, 2016; Korb et al., 2020; Mayo et al., 2018; Pawling et al., 2017; Rasch et al., 2015; Ree et al., 2019; Sato et al., 2020; Wu et al., 2015).”

28) The end paragraph of the Discussion provides clarity to some issues noted above. It would be helpful to provide this clarity early on as the comments in the early part of the Discussion section give an impression of overgeneralizing some of the findings.

We have largely rewritten the Discussion, hopefully improving clarity and avoiding overgeneralizations.

29) In the interpretation of the statistical findings please provide careful descriptions of the data and not just the p values? This includes addressing the following:a) Please go through the manuscript carefully to avoid terminology such as "did not differ", when what is meant is "did not significantly differ". It is only correct to say that two measures do not differ if they are identical.

We hope that this revised manuscript provides more careful descriptions of the data. We have changed all instances where a difference was described, now clearly stating that the difference was or was not “significantly different”. Certainly, it would be even more correct to specify each time that the differences were/weren’t “statistically significantly different”.

However, this should be fairly clear to the readers of *eLife* (especially since the wording appears most of the time in proximity of statistical values such as p values), and was skipped to facilitate text understanding.

b) It is recommended to moderate the statement that an influence of drug via mood can be excluded.

We have removed that sentence.

We now introduce the initial group comparisons with the sentence “In order to rule out eventual group differences that were not of interest, we conducted a series of statistical tests to verify the matching of the three groups.”

The presence or absence of significant differences between groups on mood and other measures is then prescribed in detail.

The passage about drug effects on mood now reads: “The three groups of participants did not differ significantly in their maximum voluntary contraction (MVC) of the hand dynamometer, which was measured right before the main task and at the end of the main task, nor in their positive and negative mood measured with the PANAS at time of pill intake and three hours later (all b < 0.6, all t < 0.8, all p > 0.4; see Table 1).”

c) P values above the α level are not accurately described as "trends"; instead please describe the pattern of the data and state clearly that the required significance level was not reached.

All p values above.05 are described as trends or as marginally significant, as long as they are also below.1. When statistically non-significant results are found, we clearly state so.

d) “Importantly, this finding cannot be explained by mouth movements that might have occurred during food delivery (although instruction to swallow followed the Delivery window), as statistics did not reveal an interaction with the factor Condition.”, a null effect (interaction) is interpreted as evidence that a potential confound did not influence the results. This is inappropriate with frequentist statistics.

We totally agree and have reformulated the sentence now stating that the absence of the interaction makes an influence unlikely: “this finding is unlikely to be explained by mouth movements related to food ingestion, because participants were instructed to swallow the food after the Delivery window, and no interaction with the factor Reward Type was found.”

30) Please replace the CS and ZM abbreviations with corrugator and zygomaticus. CS is commonly used to denote conditioned stimulus thus causes some confusion, especially when used in the graphs. Further, RewardType gives the impression that it denotes drink versus touch, rather than reward level. In Discussion paragraph six, it is suggested to clarify the announcement of possible reward or similar, since participants only received the announced rewards they chose to work for. Further, please edit phrases including "the fact that" when describing the results.

We have replaced the abbreviations CS and ZM with the words corrugator and zygomaticus (both in text and figures).

RewardType was changed to Reward Level in both text and figures.

Condition was changed to Reward Type.

In the Discussion we now distinguish between anticipation and consumption of reward.

We have removed “the fact that” from the manuscript.

31) Please provide acknowledgement of limitations in the study, which certainly exists e.g. cross-sectional design (rather than within-subjects). This can be noted when discussing the interpretation of the findings.

We have added a larger section to the Discussion, acknowledging and discussing a series of imitations, including the use of a cross-sectional design:

”The current study is characterised by a number of limitations. First, only two types of stimuli (food and touch) were used to define social and non-social rewards, and only two neurochemical systems (dopaminergic and opioidergic) were challenged. […] This reveals the difficulty of uncovering the neurochemical basis of reward processing in humans, and suggests that larger sample sizes should be used in future pharmacological studies to investigate the neurochemical bases of touch and other rewards.”

[Editors' note: further revisions were suggested prior to acceptance, as described below.]

Revisions:The authors were very responsive and provided a thorough revision of their manuscript, which is a very solid and valuable contribution to the literature. However, there are still some concerns that should be addressed.1) The corrugator and zygomaticus data is a strength of the manuscript, but the current manuscript emphasizes facial responses more than is warranted by the study design and the hypotheses. Please address the following:a) Please specify the lack of significant reductions in subjective ratings of chocolate milk and stroking touch in the Abstract.

To specify the null-effects of Drug on ratings of wanting and liking, we have added this sentence to the Abstract:

“Subjective ratings of wanting and liking were not modulated by either drug.”

b) The phrasing in the Introduction still appears to indicate that animal studies provide the best answers, whereas human subjective report is less valuable. However, the animal studies are severely hampered by their use of an indirect measure of liking, which has not been generalized beyond food reward but instead their strength in animal models is the ability to manipulate specific circuits/neurons. Please edit the text to reflect these limitations and strengths in animal models

In the Introduction we have added the following text: “In spite of the progress made, the animal research is only partly informative to comprehend reward processing in humans. While animal research allows to investigate the activity of neurons and neurotransmitters in a much more targeted way, it is also limited to certain measures of liking (i.e. behaviour and facial expressions, while humans can also provide subjective reports), and has mainly focused on food rewards in the past.”

c) The numerous figures of fEMG data patterns do not seem warranted, i.e. do they present the answers to the most important questions? It seems as though all effects below p=0.1 are illustrated.

Our intention was to show the EMG in relation to ratings of wanting/liking and effort, even when no drug effects were found. The reader might indeed be interested in seeing that the corrugator and zygomaticus activation/relaxation patterns follow expected patterns, with greater wanting and effort resulting in corrugator relaxation, and greater liking resulting in zygomaticus activation.

However, following the reviewer’s request, we have now removed former Figures 5 (Post-Effort anticipation) and 7 (Relax), which showed EMG effects with .5 > p <.1.

d) The comparable subjective wanting and liking ratings in the placebo, naltrexone and amisulpride are a main result and should be listed as such in the Abstract, Results and Discussion. Further, please clarify that face muscle responses to high-reward stimuli were also comparable after drugs. Perhaps equivalence testing or Bayesian stats could be used to address the credibility of these interesting null results?

To address the credibility of the null results on ratings, we ran Bayesian LMMs for the ratings of wanting and liking, using the package brms in R (Bürkner, 2017), and also compared the full models to identical models without the predictor Drug, using leave-one-out cross-validation (LOO-CV) based on the posterior likelihoods (Vehtari et al., 2017). This confirmed the data to be more likely under the model without the predictor Drug (weights based on averaging via stacking of predictive distributions were 0.67 and 0.998 for wanting and liking) than under the full model (weights 0.33 and 0.002 for wanting and liking). These Bayesian analyses strengthen the view, already conveyed by the frequentist LMMs, that neither drug affected wanting or liking of both types of rewards in this study. We have modified the Abstract, Results and Discussion sections, accordingly. The corresponding R scripts are available on OSF, and the model result tables have been added to the supplementary materials.

It is unclear to us, what effect of muscle responses to high-reward stimuli the reviewer is referring to.

e) Referring to reduced activity of a frowning muscle as an indication of “hedonic anticipatory pleasure” is unwarranted and likely to confuse readers with respect to what the data shows. The corrugator is not a good measure of pleasure. Similarly, the discussion of drug effects on “hedonic responses” to touch gives the impression that the drugs altered touch pleasantness, instead of a face muscle response, which was not measured in most other drug studies.

We respectfully disagree with the reviewer’s comment that “the corrugator is not a good measure of pleasure”, based on the fact that many past studies investigating hedonic responses to different types of rewards have found significant relaxation of the corrugator to more pleasant rewards, but, importantly, no reliable changes in zygomaticus activation.

This is especially the case for studies on affective touch, as in Bershad et al., 2019, Mayo et al., 2018, and Ree et al., 2019.

Studies that have administered food rewards also found effects that were either exclusively associated with, or stronger/more stable in, the corrugator compared to the zygomaticus muscle (Rasch et al., 2015; Sato et al., 2020).

Our own previous study, which used the same paradigm and the same food and touch stimuli (Korb et al., 2020), also found significant pleasure responses during anticipation and consumption of rewards only in the corrugator, but not in the zygomaticus muscle.

Finally, a study by Larsen et al., 2003, looked at corrugator and zygomaticus responses to sounds, pictures, and words representing a large range in valence. They found that: “not only that very negative and mildly negative pictures both potentiated activity over corrugator supercilii, but also that very positive pictures significantly diminished activity”, and that “even when negative affect is held constant at minimal levels, positive affect inhibits activity over corrugator supercilii”. They also report that “valence had a substantially stronger effect on activity over corrugator supercilii. In addition, more participants showed a reliable linear effect of valence on activity over corrugator supercilii.” The zygomaticus major muscle, on the other hand, shows a more complex relationship to stimulus valence, as “not only very positive, but also mildly positive and very negative pictures elicited greater activity over zygomaticus major than did neutral pictures”.

We conclude that corrugator relaxation seems, on the basis of the literature including our own previous work, a good measure of pleasure, and possibly a more reliable measure of hedonic responses during the consumption (and anticipation) of rewarding stimuli, compared to the zygomaticus muscle.

If we are missing some relevant information/published papers, we would be very grateful if the reviewer could indicate them. Unfortunately, the reference that was suggested in the previous review round (Bremhorst et al., 2019) does not seem – at least to our understanding – very relevant to this regard, as it concerns FACS coding in dogs in response to frustration, and the corrugator muscle is never mentioned.

As to the discussion of drug effects on “hedonic responses” to touch, we would like to clarify that facial responses to rewards are typically considered the “gold standard” in animal research on the neurochemical regulation of reward processing, and that it was an explicit goal of this project to measure implicit facial reactions to rewards in humans, in addition to explicit ratings of wanting/liking. We consider changes in facial EMG to be a proxy of hedonic responses (as also indicated in the manuscript), and to add valuable information complementing explicit ratings of pleasantness. Indeed, as our results (as well as other null effect findings of pharmacological manipulations on pleasantness of affective touch) suggest, pharmacological manipulation of the dopamine and/or opioid system in humans may only result in subtle effects on anticipatory and consummatory hedonic reactions, which can be seen in the type of measures taken in animal research (physical effort and facial responses) but not in the subjective ratings that are more frequently used in human research.

2) The presentation of the behavioral wanting data to illustrate the two significant two-way interactions is confusing. It's better to depict non-pooled data for each reward type and level, and use lines to indicate the significant interactions. It is challenging to determine from the figure whether there was an effect of drug for wanting of gentle caresses, for instance. Please address the following:a) The authors now provide this information in the supplement. However, showing one chart per drug instead of one per reward type/level makes it hard to discern any drug-related differences. The x axis labeling in the supplementary file was also confusing. It is recommended to remove Figure 3 and replace it with the reorganized supplementary to clarify any drug differences.

Figure 3—figure supplement 1 now shows one chart per Reward Type (instead of one chart per Drug). As it remains difficult to flag the effects of Drug X Reward Type and Drug X Reward Level, which were found for Effort, we would like to keep this figure as a supplement to Figure 3.

3) Please change the terminology to accurately depict the data and statisticsa) If the authors use 0.05 as their α level it is incorrect to describe results above 0.05 as trends or marginal effects. The higher force levels should be described as non-significantly higher. Similarly, the legend of Figure 5 makes too strong a case given the data. 1. This terminology is statistically inappropriate and maintains some unnecessary ambiguity in the literature. Many experts in statistics have pointed out this problem in recent years (see e.g. Gibbs and Gibbs 2015, https://academic.oup.com/bja/article/115/3/337/312358). It is encouraged that the authors remove these terms

We have replaced all prior mentioning of “marginally significant” or “trends for” effects, with “non-significant”.

b) It is also recommended to use more accurate descriptors of group differences, given the between-groups design, i.e. use “higher” instead of “increased”, “lower” instead of “reduced” etc.

We have changed these terms accordingly.

4) The Results section is very hard to read especially since there are multiple periods of time analyzed. Please explain in the beginning of each section what information should be gained from the analyzed data rather than providing some of this information is in the summary of EMG data. This would help the readers to better understand the data.

To improve readability of the Results section, we have summarised the main behavioural findings at the beginning of the section “Explicit measures: ratings of wanting, ratings of liking and physical effort”, and we have summarised the main EMG findings at the beginning of the section “Implicit measures: facial EMG”. We have also removed two figures, to shorten the Results section.

5) 50 mg naltrexone is not a low drug dose. It is the standard clinical dose and blocks 90-100% of mu opioid receptors. It is recommended that the authors revisit the PET literature and take into account that 50 mg naltrexone is considered sufficient to induce so-called “full” blockade of mu receptors, when interpreting their results.

We are aware that 50mg of naltrexone blocks 90% of mu-opioid receptors, and have indicated this in the Materials and methods section: “up to 90% of mu-opioid receptors in the brain remain blocked by naltrexone after 48 hours, and partial receptor blockade could be shown up to 168 hours after intake (Lee et al., 1988).”

We have now slightly changed the paragraph in the Discussion, to only say that 400mg of amisulpride may be a low dose:

“Second, we used a relatively low dose of amisulpride (400mg). […] Upon availability of drug compounds that more strongly modulate the dopaminergic systems with minimal side effects, future studies should, however, explore dose dependent changes in human reward processing.”

References

Atlas, L. Y., Wielgosz, J., Whittington, R. A., & Wager, T. D. (2014). Specifying the non-specific factors underlying opioid analgesia: Expectancy, attention, and affect. *Psychopharmacology*, *231*(5), 813–823. https://doi.org/10.1007/s00213-013-3296-1

Barr, D. J., Levy, R., Scheepers, C., & Tily, H. J. (2013). Random effects structure for confirmatory hypothesis testing: Keep it maximal. *Journal of Memory and Language*, *68*(3). https://doi.org/10.1016/j.jml.2012.11.001

Benjamini, Y., & Hochberg, Y. (1995). Controlling the False Discovery Rate: A Practical and Powerful Approach to Multiple Testing. *Journal of the Royal Statistical Society. Series B (Methodological)*, *57*(1), 289–300. JSTOR.

Bershad, A. K., Mayo, L. M., Van Hedger, K., McGlone, F., Walker, S. C., & de Wit, H. (2019). Effects of MDMA on attention to positive social cues and pleasantness of affective touch. *Neuropsychopharmacology*, *44*(10), 1698–1705. https://doi.org/10.1038/s41386-019-0402-z

Chelnokova, O., Laeng, B., Eikemo, M., Riegels, J., Loseth, G., Maurud, H., Willoch, F., & Leknes, S. (2014). Rewards of beauty: The opioid system mediates social motivation in humans. *Molecular Psychiatry*, *19*(7), 746–747. https://doi.org/10.1038/mp.2014.1

de Wit, H., & Bershad, A. K. (2020). MDMA enhances pleasantness of affective touch. *Neuropsychopharmacology*, *45*(1), 217–239. https://doi.org/10.1038/s41386-019-0473-x

Eikemo, M., Biele, G., Willoch, F., Thomsen, L., & Leknes, S. (2017). Opioid Modulation of Value-Based Decision-Making in Healthy Humans. *Neuropsychopharmacology*, *42*(9), 1833–1840. https://doi.org/10.1038/npp.2017.58

Eikemo, M., Loseth, G. E., Johnstone, T., Gjerstad, J., Willoch, F., & Leknes, S. (2016). Sweet taste pleasantness is modulated by morphine and naltrexone. *Psychopharmacology*, *233*(21–22), 3711–3723. https://doi.org/10.1007/s00213-016-4403-x

Korb, S., Massaccesi, C., Gartus, A., Lundstrom, J. N., Rumiati, R., Eisenegger, C., & Silani, G. (2020). Facial responses of adult humans during the anticipation and consumption of touch and food rewards. *Cognition*, *194*, 104044. https://doi.org/10.1016/j.cognition.2019.104044

Lang, P. J., Greenwald, M. K., Bradley, M. M., & Hamm, A. O. (1993). Looking at pictures: Affective, facial, visceral, and behavioral reactions. *Psychophysiology*, *30*(3), 261–273. Scopus. https://doi.org/10.1111/j.1469-8986.1993.tb03352.x

Larsen, J. T., Norris, C. J., & Cacioppo, J. T. (2003). Effects of positive and negative affect on electromyographic activity over zygomaticus major and corrugator supercilii. *Psychophysiology*, *40*(5), 776–785. https://doi.org/10.1111/1469-8986.00078

Lee, M. C., Wagner, H. N., Tanada, S., Frost, J. J., Bice, A. N., & Dannals, R. F. (1988). Duration of occupancy of opiate receptors by naltrexone. *Journal of Nuclear Medicine: Official Publication, Society of Nuclear Medicine*, *29*(7), 1207–1211.

Lopez-Persem, A., Rigoux, L., Bourgeois-Gironde, S., Daunizeau, J., & Pessiglione, M. (2017). Choose, rate or squeeze: Comparison of economic value functions elicited by different behavioral tasks. *PLOS Computational Biology*, *13*(11), e1005848. https://doi.org/10.1371/journal.pcbi.1005848

Matuschek, H., Kliegl, R., Vasishth, S., Baayen, H., & Bates, D. (2017). Balancing Type I error and power in linear mixed models. *Journal of Memory and Language*, *94*, 305–315. https://doi.org/10.1016/j.jml.2017.01.001

Meyer, M. C., Straughn, A. B., Lo, M. W., Schary, W. L., & Whitney, C. C. (1984). Bioequivalence, dose-proportionality, and pharmacokinetics of naltrexone after oral administration. *The Journal of Clinical Psychiatry*, *45*(9 Pt 2), 15–19.

Rosenzweig, P., Canal, M., Patat, A., Bergougnan, L., Zieleniuk, I., & Bianchetti, G. (2002). A review of the pharmacokinetics, tolerability and pharmacodynamics of amisulpride in healthy volunteers. *Human Psychopharmacology*, *17*(1), 1–13. https://doi.org/10.1002/hup.320

Weber, S. C., Beck-Schimmer, B., Kajdi, M.-E., Muller, D., Tobler, P. N., & Quednow, B. B. (2016). Dopamine D2/3- and μ-opioid receptor antagonists reduce cue-induced responding and reward impulsivity in humans. *Translational Psychiatry*, *6*(7), e850. https://doi.org/10.1038/tp.2016.113